# Natural Hypergradient Descent:
# Algorithm Design, Convergence Analysis, and Parallel Implementation

**Deyi Kong** [1]  **Zaiwei Chen** [2]  **Shuzhong Zhang** [1]  **Shancong Mou** [1]

## Abstract

In this work, we propose *Natural Hypergradient Descent* (NHGD), a new method for solving bilevel optimization problems. To address the computational bottleneck in hypergradient estimation, namely the need to compute or approximate Hessian inverses, we exploit the statistical structure of the inner optimization problem and use the empirical Fisher information matrix as an asymptotically consistent surrogate for the Hessian. This design enables a parallel *optimize-and-approximate* framework in which the Hessian-inverse approximation is updated *synchronously* with the stochastic inner optimization, reusing gradient information at negligible additional cost. Our main theoretical contribution establishes high-probability error bounds and sample complexity guarantees for NHGD that match those of state-of-the-art optimize-then-approximate methods, while significantly reducing computational time overhead. Empirical evaluations on representative bilevel learning tasks further demonstrate the practical advantages of NHGD, highlighting its scalability and effectiveness in large-scale machine learning settings.

## 1. Introduction

Bilevel optimization is an important framework in modern machine learning, with applications including hyper-data cleaning (Franceschi et al., 2017), synthetic data generation and augmentation (Mounsaveng et al., 2021; Mou et al., 2025), meta-learning (Mishchenko et al., 2023), reinforce-

---

[1]Department of Industrial and Systems Engineering, University of Minnesota, Twin Cities, Minneapolis, MN, USA [2]Edwardson School of Industrial Engineering, Purdue University, West Lafayette, IN, USA. Correspondence to: Deyi Kong <kong0280@umn.edu>, Zaiwei Chen <chen5252@purdue.edu>, Shuzhong Zhang <zhangs@umn.edu>, Shancong Mou <mou00006@umn.edu>.

*Proceedings of the 43$^{rd}$ International Conference on Machine Learning*, Seoul, South Korea. PMLR 306, 2026. Copyright 2026 by the author(s).

ment learning (Prakash et al., 2026), and physics-informed learning (Hao et al., 2023). A general bilevel optimization problem can be formulated as

$$\min_{v \in \mathbb{R}^{d_1}} \quad \Phi(v) := f(v, \theta^*(v)), \quad \text{s.t. } \theta^*(v) \in \arg\min_{\theta \in \Theta} \ell(v, \theta),$$

where $f : \mathbb{R}^{d_1} \times \mathbb{R}^{d_2} \to \mathbb{R}$ is the outer-level objective, $\ell : \mathbb{R}^{d_1} \times \mathbb{R}^{d_2} \to \mathbb{R}$ is the inner-level objective, and $\Theta := \{\theta \in \mathbb{R}^{d_2} \mid \|\theta\| \leq R\}$. Here we assume the existence of a sufficiently large $R$ such that every inner solution satisfies $\|\theta^*(v)\| < R, \ \forall v$. The constraint is therefore inactive at every $\theta^*(v)$, and the hypergradient can be derived via implicit differentiation.

*Hypergradient descent* (Ghadimi & Wang, 2018) is widely used for solving bilevel optimization problems. Assuming the inner-level problem admits a unique minimizer $\theta^*(v)$, the hypergradient is given by

$$\nabla \Phi(v) = \nabla_v f(v, \theta^*(v)) + \nabla \theta^*(v)^\top \nabla_\theta f(v, \theta^*(v)). \quad (1)$$

Under mild regularity conditions, the implicit function theorem (Krantz & Parks, 2002) yields

$$\nabla^2_{\theta,v} \ell(v, \theta^*(v)) + H(\theta^*(v)) \nabla \theta^*(v) = 0, \quad (2)$$

where $H(\theta^*(v)) := \nabla^2_{\theta,\theta} \ell(v, \theta^*(v))$ denotes the (invertible) Hessian of the inner objective. Eq. (1) and Eq. (2) yield a closed-form expression for the hypergradient:

$$\nabla \Phi(v) = \nabla_v f(v, \theta^*(v)) \\ \quad - \left(\nabla^2_{\theta,v} \ell(v, \theta^*(v))\right)^\top H(\theta^*(v))^{-1} \nabla_\theta f(v, \theta^*(v)).$$

Directly evaluating the Hessian inverse $H(\theta^*(v))^{-1}$ is prohibitively expensive for large-scale inner problems. As a result, most existing hypergradient methods follow an *optimize-then-approximate* paradigm: the inner problem is first (approximately) solved to near-optimality, after which the Hessian inverse is approximated *post hoc* using numerical or algorithmic techniques such as Neumann series expansions (Ghadimi & Wang, 2018; Lorraine et al., 2020; Ji et al., 2021), conjugate gradient methods (Pedregosa, 2016; Rajeswaran et al., 2019; Yang et al., 2023), quadratic solvers (Arbel & Mairal, 2022), or fixed-point iterations (Grazzi et al., 2020). However, this post hoc Hessian

inverse approximation is inherently sequential and can remain computationally expensive, even when using efficient solvers. This raises a natural question:

*Can we do better by exploiting additional structure in inner-level optimization problems?*

In this work, we provide an affirmative answer by focusing on an important class of bilevel optimization problems in which the inner objective corresponds to Kullback–Leibler (KL) divergence minimization—a ubiquitous setting in machine learning (Kingma & Welling, 2014; Murphy, 2022), with the following loss function:

$$\ell(v, \theta) := \mathbb{E}_{\xi \sim q(\cdot)}[l(v, \theta, \xi)], \ l(v, \theta, \xi) := -\log p(\xi; v, \theta),$$

where $\xi$ is drawn from the data distribution $q(\cdot)$ and $p(\cdot; v, \theta)$ denotes the model distribution parameterized by $(v, \theta)$.

Under this setting, we adopt a fundamentally different perspective by exploiting the *statistical structure* of the inner optimization problem. To this end, we recall two key quantities that characterize the statistical structure of inner-level objective: the Fisher information matrix (FIM) $I(\theta) := \mathbb{E}_{\xi \sim q(\cdot)}\left[\nabla_\theta l(v, \theta, \xi) \nabla_\theta l(v, \theta, \xi)^\top\right]$ and the empirical Fisher information matrix (EFIM) $\hat{I}_t := \frac{1}{t} \sum_{i=0}^{t-1} \nabla_\theta l(v, \theta, \xi_i) \nabla_\theta l(v, \theta, \xi_i)^\top$. Rather than relying on numerical or algorithmic approximations, we approximate the Hessian inverse asymptotically using the EFIM inverse. Crucially, the EFIM can be *synchronously* updated alongside the stochastic inner-level optimization by reusing stochastic gradients at each iteration. This enables a parallel *optimize-and-approximate* paradigm that fundamentally differs from existing optimize-then-approximate methods, and avoids the computational time overhead associated with post hoc Hessian inverse estimation.

In single-level optimization, using the FIM inverse as a surrogate for the Hessian inverse naturally leads to *Natural Gradient Descent* (Amari, 1998) . Despite its success, this idea has been largely unexplored in bilevel optimization. We term our approach *Natural Hypergradient Descent* (NHGD; see Figure 1), and show that this approximation is in fact even *more natural* in the bilevel setting, leading to favorable theoretical properties and significant practical advantages.

**Contributions.** Our contributions are as follows:

1. *Practically*, we propose a novel perspective for approximating the Hessian inverse in bilevel optimization using the EFIM inverse. This perspective offers several practical benefits: (1) it avoids explicit Hessian construction; (2) it reuses stochastic gradients from the inner-level stochastic gradient descent (SGD) at negligible additional cost; (3) it enables parallel hypergradient computation during inner-level optimization, eliminating post hoc Hessian inverse

approximation overhead; (4) it admits further acceleration via K-FAC (Martens & Grosse, 2015) for large-scale deep learning problems. 2. *Theoretically*, we establish high-probability convergence guarantees for NHGD: (1) a high-probability sample complexity bound for the Hessian inverse approximation (Theorem 4.8); and (2) a high-probability finite-sample convergence rate for the outer-level objective to reach an $\epsilon$-stationary point (Theorem 4.11).

Overall, the NHGD matches the sample complexity of state-of-the-art *optimize-then-approximate* hypergradient methods (see Table 1), while providing improved computational efficiency in practice.

## 2. Related Works

### 2.1. Hypergradient Descent

Hypergradient descent is a widely used method for bilevel optimization. But the key computational bottleneck lies in estimating hypergradients, which requires computing or approximating the Hessian inverse of the inner objective. To address this, several methods have been proposed.

**Iterative Differentiation/Algorithm Unrolling (ITD)** unrolls the computational graph of the inner problem and then differentiate through the trajectory using automatic differentiation (AD) (Domke, 2012; Maclaurin et al., 2015; Franceschi et al., 2017; 2018). However, the computational and memory costs of ITD scale with the number of inner optimization iterations, limiting the number of unrolled iterations used in practice (Shaban et al., 2019).

**Approximate Implicit Differentiation (AID)** methods approximate the hypergradient by either explicitly approximating the Hessian inverse or approximately solving the linear system induced by the implicit differentiation condition in Eq. (2). Explicit Hessian inverse approximations typically rely on truncated Neumann series expansions (Lorraine et al., 2020; Ji et al., 2021; Hong et al., 2023). Alternatively, Eq. (2) can be approximately solved using conjugate gradient (CG) methods (Pedregosa, 2016; Rajeswaran et al., 2019; Grazzi et al., 2020; Yang et al., 2023), quadratic subproblem solvers (Grazzi et al., 2020; Arbel & Mairal, 2022), or fixed-point iterations (Grazzi et al., 2020).

From an algorithmic perspective, existing methods can be further categorized based on their iteration structure. Early approaches adopt a *double-loop* or *optimize-then-approximate* paradigm, in which the inner problem is first solved to a near-optimal point, followed by a separate hypergradient approximation step (Ghadimi & Wang, 2018; Lorraine et al., 2020; Ji et al., 2021; Arbel & Mairal, 2022). More recent studies have proposed *single-loop* methods that update the inner and outer variables simultaneously, including two-time-scale algorithms (Hong et al., 2023) and

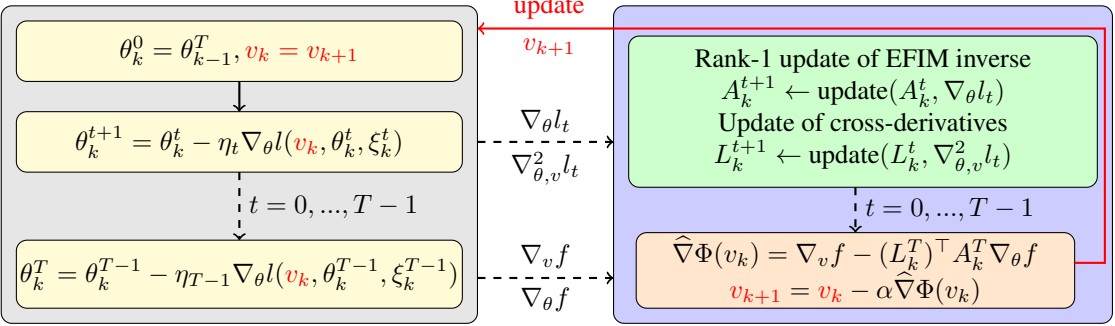

*Figure 1.* Overview of NHGD. The inner problem is solved using SGD on Device 1, while gradient information is sent to Device 2 for iterative rank-one updates of the EFIM inverse $A_t$ and the cross-derivatives $L_t$. After the inner loop, Device 1 sends gradient information to Device 2, which approximates the hypergradient and updates the outer variable. The updated $v_{k+1}$ is then returned to Device 1 to resume the inner optimization from $\theta_k^t$. This design enables synchronous hypergradient estimation alongside inner-loop optimization.

*Table 1.* Summary of stochastic bilevel algorithms. **Sample Complexity** counts the total number of samples used for estimating gradients and Jacobian(Hessian)–vector products. Since the outer-level problem is deterministic in our setting, baseline complexities are recomputed accordingly. The $\tilde{\mathcal{O}}$ notation omits the $\log \frac{1}{\epsilon}$ term. **MSE** and **H.P.** denote mean-square bounds and high-probability bounds, respectively. For Hessian inversion, **Neumann** indicates truncated Neumann series approximation, while **SGD** solves the implicit linear system via SGD. **Serial/Parallel** specify whether inner-level optimization and hypergradient approximation can run in parallel. We omit the comparison with variance reduction-based methods: VRBO, MRBO (Yang et al., 2021); FLSA (Li et al., 2022), etc.

| Algorithm Structure | Method | Sample Complexity | Bound Type | Hessian Inversion | Batch Size |
|---|---|---|---|---|---|
| Double Loop | BSA (Ghadimi & Wang, 2018) | $\tilde{\mathcal{O}}(\epsilon^{-3})$ | MSE | Neumann(Serial) | $\tilde{\mathcal{O}}(1)$ |
| | stocBiO (Ji et al., 2021) | $\tilde{\mathcal{O}}(\epsilon^{-2})$ | MSE | Neumann(Serial) | $\tilde{\mathcal{O}}(\epsilon^{-1})$ |
| | ALSET (Chen et al., 2021) | $\tilde{\mathcal{O}}(\epsilon^{-2})$ | MSE | Neumann(Serial) | $\tilde{\mathcal{O}}(1)$ |
| | AmIGO (Arbel & Mairal, 2022) | $\mathcal{O}(\epsilon^{-2})$ | MSE | SGD(Serial) | $\mathcal{O}(\epsilon^{-1})$ |
| | **NHGD (Ours)** | $\tilde{\mathcal{O}}(\epsilon^{-2})$ | H.P. | EFIM(Parallel) | $\mathcal{O}(1)$ |
| Single Loop | TTSA (Hong et al., 2023) | $\tilde{\mathcal{O}}(\epsilon^{-2.5})$ | MSE | Neumann(Serial) | $\tilde{\mathcal{O}}(1)$ |
| | SOBA (Dagréou et al., 2022) | $\mathcal{O}(\epsilon^{-2})$ | MSE | SGD(Parallel) | $\mathcal{O}(1)$ |

SOBA (Dagréou et al., 2022). Despite these advances, hypergradient estimation remains the central computational challenge, and existing methods still rely on numerical or algorithmic approximation for handling the Hessian inverse.

In this work, we take a first step toward exploring *statistical* approximations of the Hessian inverse in the bilevel optimization setting, and focus on the classical double-loop structure for clarity of analysis. While the proposed approach can be extended to single-loop variants and enhanced with variance-reduction techniques, we leave these directions for future work. A summary of representative works is provided in Table 1.

### 2.2. Other Hessian Inverse Approximation Methods

Hessian inverse approximation under structural assumptions has been widely studied. Representative approaches include low-rank approximations such as Nyström methods (Hataya & Yamada, 2023) and block-diagonal or

Kronecker-factored schemes motivated by neural network architectures (Martens & Grosse, 2015). While computationally efficient, these methods often lack statistical consistency or rigorous convergence guarantees. Quasi-Newton methods are also closely related. This branch of methods construct Hessian inverse approximations via low-rank updates satisfying the secant condition, including BFGS, DFP, SR1, and Broyden-type methods (Nocedal & Wright, 2006). While such approaches have been explored for hypergradient computation (to approximate the Hessian inverse (Hao et al., 2023)), they generally do not provide global convergence guarantees.

### 2.3. Natural Gradient Descent

Adopting the EFIM inverse to approximate the Hessian inverse is closely related to *single-level Natural Gradient Descent* (NGD) (Amari, 1998; Martens, 2020) which preconditions gradient updates using the EFIM inverse for KL-divergence minimization.

The hypergradient depends on the Hessian inverse evaluated at the inner optimum. At this point, the FIM coincides with the true Hessian of the inner-level objective (see Proposition B.2), regardless of whether the overall problem has converged to its optimum. In contrast, for NGD, the FIM generally does not coincide with the true Hessian, except at the optimum of the overall problem. Therefore, in bilevel optimization, using the FIM as a Hessian surrogate is more appropriate and appealing. Moreover, as the number of inner-level SGD iterations increases, the EFIM converges asymptotically to the FIM, and consequently to the true Hessian. This property makes EFIM-based approximations especially well suited for bilevel optimization.

## 3. Natural Hypergradient Descent Algorithm

In this section, we present the proposed NHGD algorithm, summarized in Algorithm 1. The overall algorithm follows a standard double-loop structure: the outer-loop performs gradient descent on the outer variable $v$, while the inner-loop approximately solves the inner problem using SGD algorithm. Crucially, during the inner-loop, the Hessian inverse approximation is *synchronously* updated using the EFIM inverse at each SGD step by reusing the stochastic gradients (Line 7 of Algorithm 1). This design enables efficient parallel execution of inner problem optimization and hypergradient approximation, thereby avoiding the runtime overhead associated with post hoc hypergradient approximation approaches, and constitutes the main novelty of the proposed algorithm. Below, we describe the three core components of NHGD in detail.

**SGD algorithm for inner problem:** Given the outer variable $v_k$, at each iteration of the inner-loop, we sample data $\xi_k^t \sim q(\cdot)$ and apply projected SGD:

$$\theta_k^{t+1} = \Pi_\Theta \left( \theta_k^t - \eta_t \nabla_\theta l(v_k, \theta_k^t, \xi_k^t) \right), \quad (3)$$

where the stochastic gradient $\nabla_\theta l(v_k, \theta_k^t, \xi_k^t)$ is the unbiased estimator of the true gradient $\nabla_\theta \ell(v_k, \theta_k^t)$. Under standard assumptions, $\theta_k^t$ converges to the inner solution $\theta^*(v_k)$. A detailed convergence analysis is provided in Lemma 4.9.

**Iterative approximation of the Hessian inverse:** A key challenge in hypergradient descent lies in evaluating the Hessian inverse. When the inner problem corresponds to KL-divergence minimization, the FIM coincides with the Hessian at the optimum, i.e., $H(\theta^*(v)) = I(\theta^*(v))$ (see Proposition B.2). This Hessian–FIM equivalence motivates our approximation strategy: the FIM inverse serves as an efficient surrogate for the Hessian inverse. We further approximate the FIM using the EFIM computed along the

inner optimization trajectory $\{\theta_k^0, \ldots, \theta_k^t\}$:

$$I_k^{t+1} := \frac{1}{t+1} \sum_{i=0}^{t} \nabla_\theta l(v_k, \theta_k^i, \xi_k^i) \nabla_\theta l(v_k, \theta_k^i, \xi_k^i)^\top$$

$$= \frac{t}{t+1} I_k^t + \frac{1}{t+1} \nabla_\theta \ell(v_k, \theta_k^t, \xi_k^t) \nabla_\theta \ell(v_k, \theta_k^t, \xi_k^t)^\top.$$

Intuitively, by the convergence of the inner-loop SGD and the Law of Large Numbers, the EFIM $I_k^t$ converges to the FIM $I(\theta^*(v_k))$, which in turn equals the Hessian $H(\theta^*(v_k))$ at the inner optimum.

Although the update of $I_k^t$ reuses stochastic gradients from the inner-loop SGD at negligible cost, explicitly inverting $I_k^t$ remains computationally expensive. A key observation is that each update of $I_k^t$ is a rank-one outer product, which allows its inverse to be updated efficiently via the Sherman–Morrison formula. Specifically, letting $A_k^t := (I_k^t)^{-1}$, the inverse can be updated as follows:

$$A_k^{t+1} = \frac{t+1}{t} A_k^t \quad (4)$$

$$- \frac{\frac{t+1}{t^2} A_k^t \nabla_\theta l(v_k, \theta_k^t, \xi_k^t) \left( A_k^t \nabla_\theta l(v_k, \theta_k^t, \xi_k^t) \right)^\top}{1 + \frac{1}{t} \nabla_\theta l(v_k, \theta_k^t, \xi_k^t)^\top A_k^t \nabla_\theta l(v_k, \theta_k^t, \xi_k^t)}.$$

As $I_k^t$ converges to $H(\theta^*(v_k))$, its inverse $A_k^t$ provides an efficient approximation of $H(\theta^*(v_k))^{-1}$. A non-asymptotic justification is established in Theorem 4.8.

*Remark* 3.1. The update of $A_k^t$ reuses stochastic gradients from the inner-loop SGD, making the procedure efficient and *highly parallelizable*. Specifically, stochastic gradients from inner-loop can be sent to a separate device for synchronous hypergradient updates, requiring only one-way communication during the inner-loop, as shown in Figure 1.

**Iterative approximation of the cross-partial derivative:** We approximate the cross-partial derivative $\nabla_{\theta,v}^2 \ell(v, \theta^*(v))$ using the sample average along the inner optimization trajectory $\{\theta_k^0, \cdots, \theta_k^{t-1}\}$:

$$L_k^{t+1} = \frac{1}{t+1} \sum_{i=0}^{t} \nabla_{\theta,v}^2 l(v_k, \theta_k^i, \xi_k^i)$$

$$= \frac{t}{t+1} L_k^t + \frac{1}{t+1} \nabla_{\theta,v}^2 l(v_k, \theta_k^t, \xi_k^t). \quad (5)$$

Similarly, by the convergence of inner-loop SGD and the Law of Large Numbers, the $L_k^t$ is expected to converge to the $\nabla_{\theta,v}^2 \ell(v, \theta^*(v))$. A non-asymptotic justification of this convergence is given in Proposition 4.13.

In practice, we can also approximate the cross-partial derivative using the sample average at the end of inner loop: $\hat{L}_k^m = \frac{1}{m} \sum_{i=0}^{m-1} \nabla_{\theta,v}^2 l(v_k, \theta_k^T, \xi_k^i)$. Both estimators $L_k^t$ and $\hat{L}_k^m$ involve trade-offs. The trajectory-based estimator $L_k^t$ reuses inner-loop SGD samples and computational

graphs to compute Jacobian–vector products, but requires storing the full inner-loop graph, which can be memory intensive. In contrast, $\hat{L}_k^m$ avoids retaining the inner-loop graph and is more memory efficient, at the cost of additional samples. Lemma C.3 shows that with $m = \mathcal{O}(T)$, $\hat{L}_k^m$ achieves the same non-asymptotic convergence rate as $L_k^T$.

Finally, with Eq. (4) and Eq. (5), the hypergradient can be approximated as:

$$\widehat{\nabla}\Phi(v_k) = \nabla_v f(v_k, \theta_k^T) - (L_k^T)^\top A_k^T \, \nabla_\theta f(v_k, \theta_k^T),$$

and the outer-level variable is updated as:

$$v_{k+1} = v_k - \alpha \widehat{\nabla}\Phi(v_k).$$

---

**Algorithm 1** Natural Hypergradient Descent

1: **Input:** $v_1, \theta_0^T, \alpha, \{\eta_t\}, T, K, A_0^T$ and $L_0^T$
2: **for** $k = 1, \dots, K$ **do**
3:     Initialize $\theta_k^0 = \theta_{k-1}^T$, $A_k^0 = A_{k-1}^T$ and $L_k^0 = L_{k-1}^T$
4:     **for** $t = 0, \dots, T-1$ **do**
5:         Sample data point $\xi_k^t$, and compute $\nabla_v l(v_k, \theta_k^t, \xi_k^t), \nabla_\theta l(v_k, \theta_k^t, \xi_k^t), \nabla_\theta^2 l(v_k, \theta_k^t, \xi_k^t)$
6:         Update the inner variable $\theta_k^t \to \theta_k^{t+1}$ via Eq. (3).
7:         Update the EFIM inverse $A_k^t \to A_k^{t+1}$ for the iterative Hessian inverse approximation via Eq. (4).
8:         Update the cross derivative $L_k^t \to L_k^{t+1}$ via Eq.(5).
9:     **end for**
10:     Calculate the hypergradient approximation: $\widehat{\nabla}\Phi(v_k) = \nabla_v f(v_k, \theta_k^T) - (L_k^T)^\top A_k^T \nabla_\theta f(v_k, \theta_k^T)$.
11:     Update $v_{k+1} = v_k - \alpha \widehat{\nabla}\Phi(v_k)$.
12: **end for**

---

**Practical Considerations** (1) *Synchronous and parallel hypergradient computation:* The EFIM inverse update (4) and the cross-partial derivative update (5) can both run in parallel with the inner SGD (see Figure 1). As a result, the hypergradient is approximated synchronously with the inner optimization, allowing immediate outer updates without extra runtime overhead. The formal parallel implementation of NHGD (Algorithm 2) can be found in Appendix A. (2) *Memory-Efficient Approximation via K-FAC (Martens & Grosse, 2015):* K-FAC exploits the layer-wise structure of neural networks to construct a Kronecker-factored approximation of the FIM, enabling the EFIM inverse update to be performed in a block-wise manner. Incorporating K-FAC greatly reduces memory and computational cost, enabling NHGD to scale to larger models.

## 4. Convergence Analysis

In this section, we present the convergence analysis of the proposed NHGD algorithm. The analysis is built upon two main theoretical results. Theorem 4.8 establishes the accuracy of the proposed Hessian inverse approximation, which serves as the key technical foundation of our method. Building on this result, Theorem 4.11 characterizes the overall convergence rate of NHGD.

Before presenting these two main theorems, we first introduce a set of standard assumptions commonly adopted in the analysis of bilevel optimization problems (Ghadimi & Wang, 2018; Ji et al., 2021; Arbel & Mairal, 2022; Chen et al., 2021; 2022; Hong et al., 2023).

**Assumption 4.1.** (i) $l(v, \theta, \xi)$ is $\mu$-strongly convex in $\theta$, twice continuously differentiable in $(v, \theta)$ and jointly $L$-smooth in $(v, \theta)$, uniformly for all $\xi$.

(ii) For any $v$, $\nabla_{\theta,v}^2 \ell(v, \cdot)$, $\nabla_{\theta,\theta}^2 \ell(v, \cdot)$ are $L_{\ell_{\theta,v}}$, $L_{\ell_{\theta,\theta}}$-Lipschitz continuous. For any $\theta$, $\nabla_{\theta,v}^2 \ell(\cdot, \theta)$, $\nabla_{\theta,\theta}^2 \ell(\cdot, \theta)$ are $\bar{L}_{\ell_{\theta,v}}$, $\bar{L}_{\ell_{\theta,\theta}}$-Lipschitz continuous.

**Assumption 4.2.** (i) For any $v$, $\nabla_v f(v, \cdot)$, $\nabla_\theta f(v, \cdot)$ is $L_{f_v}$, $L_{f_\theta}$-Lipschitz continuous. For any $\theta$, $\nabla_v f(\cdot, \theta)$, $\nabla_\theta f(\cdot, \theta)$ is $\bar{L}_{f_v}$, $\bar{L}_{f_\theta}$-Lipschitz continuous.

(ii) For any $v$ and $\theta$, there exists constant $D_1 > 0$ such that $\|\nabla_\theta f(v, \theta)\| \le D_1$.

**Assumption 4.3.** For any $v$ and $\theta$, the stochastic derivative $\nabla_v l(v, \theta, \xi)$, $\nabla_\theta l(v, \theta, \xi)$, $\nabla_{v,\theta}^2 l(v, \theta, \xi)$ and $\nabla_{\theta,\theta}^2 l(v, \theta, \xi)$ are unbiased estimator of $\nabla_v \ell(v, \theta)$, $\nabla_\theta \ell(v, \theta)$, $\nabla_{v,\theta}^2 \ell(v, \theta)$ and $\nabla_{\theta,\theta}^2 \ell(v, \theta)$.

Assumptions 4.1 and 4.2 imply the Lipschitz continuity of the solution mapping $\theta^*(v)$ and the smoothness of the composite outer objective $\Phi(v)$, as formalized in Lemma 2.2 of Ghadimi & Wang (2018). Since the outer objective $\Phi(v)$ is generally nonconvex, such smoothness properties are essential for establishing convergence guarantees to stationary points. Assumption 4.3 ensures unbiasedness of the stochastic first and second-order derivative estimators used in the proposed algorithm.

**Assumption 4.4.** There exist constant $C_1 > 0$ such that
$$\sup_{v \in \mathbb{R}^{d_1}, \theta \in \Theta, \xi} \|\nabla_\theta l(v, \theta, \xi) - \nabla_\theta \ell(v, \theta)\| \le C_1.$$

Assumption 4.4 is widely used in existing studies on general stochastic approximation algorithms, including, but not limited to, stochastic gradient descent (Karimi et al., 2019) and other data-driven machine learning algorithms (Bertsekas & Tsitsiklis, 1996). An interesting direction for future work is to investigate whether Assumption 4.4 can be relaxed to allow unbounded but light-tailed (e.g., sub-Gaussian) noise.

We now impose the following assumption, under which the hypergradient exists and the inner variable remains bounded. The boundedness of the inner variable is essential to our analysis: our Hessian-inverse estimation bound depends on $\|\theta_k^0 - \theta^*(v_k)\|$ in both the numerator and the denominator,

so a simple telescoping argument to bound $\|\theta_k^T - \theta^*(v_k)\|$ does not work. The term $\|\theta_k^0 - \theta^*(v_k)\|$ itself requires an a priori bound, which the compactness of $\Theta$ provides.

**Assumption 4.5.** There exists a constant $R > 0$ such that for every $v \in \mathbb{R}^{d_1}$, the inner solution $\theta^*(v)$ lies in the interior of the constraint set $\Theta := \{\theta \in \mathbb{R}^{d_2} \mid \|\theta\| \le R\}$.

Finally, we introduce a structural assumption that is crucial for enabling an efficient Hessian inverse approximation.

**Assumption 4.6.** For any $v$, the inner minimizer $\theta^*(v)$ yields a model distribution that matches the true data distribution $q(\cdot)$, i.e., $p(\cdot; v, \theta^*(v)) = q(\cdot)$.

This assumption guarantees the equivalence between the Hessian of the inner objective and the FIM evaluated at the inner-level optimum, which is critical for our Hessian inverse approximation strategy.

*Remark* 4.7. In practice, exact model specification may not hold. A natural direction for future work is to extend the analysis to the misspecified setting in which

$$\text{TV}(p(\cdot; v, \theta^*(v)), q(\cdot)) \le \epsilon_D \quad \text{for all } v,$$

where $\text{TV}(\cdot, \cdot)$ denotes total variation distance.

### 4.1. Analysis for Hessian Inverse Approximation Bound

In this section, we establish a high-probability sample complexity bound showing that the *EFIM inverse* converges to the true *Hessian inverse at the inner optimum*. This result is a key component of the convergence analysis of NHGD. We first state the result and then outline the proof sketch.

#### 4.1.1. HESSIAN INVERSE APPROXIMATION BOUND

**Theorem 4.8.** *Suppose Assumptions 4.1, 4.3, 4.4 and 4.6 hold. For any outer iteration $k$, given the outer variable $v_k$, $\theta_k^0$ and set the inner stepsize as $\eta_t = 4/(\mu(t + \frac{8L^2}{\mu^2}))$. For any $\delta \in (0, 1)$ and $T \ge T_0(\delta)$, the following holds with probability at least $1 - \delta$:*

$$\left\|A_k^T - H(\theta^*(v_k))^{-1}\right\| = \mathcal{O}\left(\frac{1}{\sqrt{T}}\sqrt{1 + \log\left(\frac{1+T}{\delta}\right)}\right),$$

*where $T_0(\delta)$ is defined in* (21).

Our analysis establishes convergence guarantee in a high-probability sense. In contrast, prior works such as (Ghadimi & Wang, 2018; Ji et al., 2021; Arbel & Mairal, 2022; Chen et al., 2021; 2022; Hong et al., 2023) analyze convergence using expectation-based metrics. Consequently, our result is stronger in the sense that a high-probability bound with a light tail can be directly translated into a mean bound via the identity $\mathbb{E}[X] = \int_0^\infty \mathbb{P}(X > x)\, dx$ for any non-negative random variable $X$. By contrast, starting from a mean bound, standard tools such as Markov or Chebyshev inequalities typically yield only high-probability bounds with polynomial (power-law) tails.

#### 4.1.2. PROOF SKETCH OF THEOREM 4.8

We now outline the key steps and intermediate results used to establish the Hessian inverse approximation bound.

**Step 1: Reduction to EFIM error.** We aim to bound the error $\|A_k^T - H(\theta^*(v_k))^{-1}\|$, or equivalently $\|(I_k^T)^{-1} - I(\theta^*(v_k))^{-1}\|$, where the equivalence follows from $I(\theta^*(v_k)) = H(\theta^*(v_k))$ (stated in Proposition B.2).

Under the $\mu$-strong convexity of the inner objective $\ell$ and by Weyl's eigenvalue inequality, we further have

$$\left\|(I_k^T)^{-1} - I(\theta^*(v_k))^{-1}\right\| \le \frac{\|I_k^T - I(\theta^*(v_k))\|}{(\mu - \|I_k^T - I(\theta^*(v_k))\|)\mu}.$$

Therefore, controlling the Hessian inverse approximation error reduces to bounding the error $\|I_k^T - I(\theta^*(v_k))\|$.

**Step 2: Decomposition of the EFIM error.** The main challenge of bounding $\|I_k^T - I(\theta^*(v_k))\|$ lies in the fact that $I_k^T$ is computed along the trajectory $\{\theta_k^0, \ldots, \theta_k^{T-1}\}$, whereas $I(\theta^*(v_k))$ is defined at the inner-level optimum. Let $G_k^t := \nabla_\theta l(\theta_k^t, \xi_k^t)\nabla_\theta l(\theta_k^t, \xi_k^t)^\top$, the EFIM error can be decomposed as:

$$\|I_k^T - I(\theta^*(v_k))\|$$
$$\le \frac{1}{T}\underbrace{\left\|\sum_{t=0}^{T-1}\left(G_k^t - \mathbb{E}_t[G_k^t]\right)\right\|}_{\text{stochastic error}} + \frac{1}{T}\underbrace{\left\|\sum_{t=0}^{T-1}\left(\mathbb{E}_t[G_k^t] - I(\theta^*(v_k))\right)\right\|}_{\text{optimization error}},$$

where $\mathbb{E}_t[\cdot] := \mathbb{E}[\cdot \mid \sigma(v_k, \theta_k^0, \xi_k^0, \ldots, \xi_k^{t-1})]$ and the $\sigma\{\cdot\}$ denotes the $\sigma$-algebra generated by the random variables.

The stochastic error term captures the deviation introduced by stochastic sampling, while the optimization error quantifies the deviation of $\theta_k^t$ from the inner optimum $\theta^*(v_k)$.

**Step 3: Bounding stochastic and optimization errors.** The stochastic error is controlled via the matrix Azuma inequality. The optimization error depends on the convergence of the inner-level SGD. Under the smoothness and gradient noise assumptions for $l$, controlling the optimization error reduces to bounding $\sum_{t=0}^{T-1} \|\theta_k^t - \theta^*(v_k)\|$. Therefore, we establish a high-probability bound on the inner-level SGD trajectory in the following lemma.

**Lemma 4.9.** *Suppose Assumptions 4.1, 4.3 and 4.4 hold. For any outer iteration $k$, given the outer variable $v_k$, $\theta_k^0$ and set the inner stepsize as $\eta_t = 4/(\mu(t+q))$ and $q \ge 8L^2/\mu^2$. Then, for any $\delta \in (0, 1)$, with probability at least $1 - \delta$,*

$$\|\theta_k^t - \theta^*(v_k)\|^2 \le \left(\frac{q}{t+q}\right)^2 \|\theta_k^0 - \theta^*(v_k)\|^2 + \frac{c'\log(e/\delta)}{t+q},$$

*where $c'$ is the constant defined in* (16).

*Remark* 4.10. The proof of Lemma 4.9 relies on bounding the moment-generating function of $\|\theta_k^t - \theta^*(v_k)\|^2$, which yields a significantly tighter *high-probability* control than the mean-square bound $\mathbb{E}\left[\|\theta_k^t - \theta^*(v_k)\|^2\right]$. The detailed proof is provided in Appendix C.1.

Combining the bounds on the stochastic and optimization errors yields a high-probability bound on $\|I_k^T - I(\theta^*(v_k))\|$, and consequently the desired Hessian inverse approximation error, which completes the proof of Theorem 4.8. Detailed derivations are deferred to Appendix C.3.

### 4.2. Analysis for NHGD Convergence Rate

In this section, building on the Hessian inverse approximation result established in Section 4.1.1, we characterize the convergence of the NHGD algorithm. We present the main convergence theorem and then outline the proof sketch.

#### 4.2.1. CONVERGENCE RATE

**Theorem 4.11.** *Suppose Assumptions 4.1, 4.2, 4.3, 4.4 and 4.6 hold. Set the parameters for Algorithm 1 as $\alpha_k = 1/(4L_v)$ and $\eta_t = 4/(\mu(t + \frac{8L^2}{\mu^2}))$, for any $\delta \in (0,1)$ and $T \geq T_1(\delta)$, the following holds with probability at least $1 - \delta$:*

$$\frac{1}{K} \sum_{k=0}^{K-1} \|\nabla \Phi(v_k)\|^2 = \mathcal{O}\left(\frac{1}{K} + \frac{1}{T} \log\left(\frac{KT}{\delta}\right)\right),$$

*where $L_v$ is defined in (22) and $T_1(\delta)$ is defined in (44).*

*Equivalently, by choosing $T = \mathcal{O}(\epsilon^{-1} \log(\delta^{-1}\epsilon^{-1}))$ and $K = \mathcal{O}(\epsilon^{-1})$, NHGD achieves an $\epsilon$-stationary point, with overall sample complexity $\tilde{\mathcal{O}}(\epsilon^{-2})$.*

*Remark* 4.12. NHGD achieves an $\epsilon$-stationary point with total sample complexity $\tilde{\mathcal{O}}(\epsilon^{-2})$, matching that of optimize-then-approximate methods (see Table 1). Notably, the Hessian inverse approximation in NHGD can be performed in parallel with the inner-loop SGD, incurring no additional runtime overhead. This computational efficiency is further validated by the numerical results in Section 5.

#### 4.2.2. PROOF SKETCH OF THEOREM 4.11

We now outline the main steps in the convergence analysis of the outer-level iterates.

**Step 1: Descent inequality for the outer level.** We begin by establishing a descent inequality for the outer-level objective. Under Assumptions 4.1 and 4.2 , the outer iterates satisfy $\sum_{k=1}^K \|\nabla \Phi(v_k)\|^2 \leq 16L_v(\Phi(v_1) - \Phi^*) + 3 \sum_{k=1}^K \|\nabla \Phi(v_k) - \widehat{\nabla} \Phi(v_k)\|^2$, where $\Phi^*$ denotes the optimal value of the outer objective. A formal statement of this result is provided in Lemma D.1. This shows that establishing convergence guarantee reduces to bounding the hypergradient approximation error $\|\nabla \Phi(v_k) - \widehat{\nabla} \Phi(v_k)\|$.

**Step 2: Decomposition of the hypergradient approximation error.** Under the regularity conditions imposed by our assumptions, the hypergradient approximation error admits the following decomposition:

$$\|\nabla \Phi(v_k) - \widehat{\nabla} \Phi(v_k)\|$$
$$\leq \mathcal{O}\left(\|\theta_k^T - \theta^*(v_k)\|\right) + \mathcal{O}\left(\|A_k^T - H(\theta^*(v_k))^{-1}\|\right)$$
$$+ \mathcal{O}\left(\|(L_k^T - \nabla_{\theta,v}^2 \ell(v_k, \theta^*(v_k)))A_k^T\|\right),$$

where the formal statement of this result is provided in Lemma D.2. Consequently, controlling the hypergradient error reduces to bound: (i) the inner-level SGD deviation $\|\theta_k^T - \theta^*(v_k)\|$, (ii) the Hessian inverse approximation error $\|A_k^T - H(\theta^*(v_k))^{-1}\|$, and (iii) the cross-partial derivative approximation error $\|L_k^T - \nabla_{\theta,v}^2 \ell(v_k, \theta^*(v_k))\|$.

We next bound the inner-level SGD deviation $\|\theta_k^T - \theta^*(v_k)\|$, whose error is coupled with $\|\nabla \Phi(v_k)\|$, reflecting the intrinsic coupling between the inner-level SGD deviation and the outer objective. The Hessian inverse and cross-partial approximation errors, however, admit bounds independent of $v_k$, and are therefore addressed separately at the end.

**Step 3: Bounding the inner-level SGD deviation.** Lemma 4.9 shows that inner-level SGD error depends on $\|\theta_k^0 - \theta^*(v_k)\|$. To further control this, we use the warm-start technique $\theta_k^0 = \theta_{k-1}^T$, which yields the following recursion: $\|\theta_{k-1}^T - \theta^*(v_k)\| \leq \|\theta_{k-1}^T - \theta^*(v_{k-1})\| + \frac{\alpha L}{\mu} \|\widehat{\nabla} \Phi(v_{k-1}) - \nabla \Phi(v_{k-1})\| + \frac{\alpha L}{\mu} \|\nabla \Phi(v_{k-1})\|$. Applying this recursion yields a high-probability control of the hypergradient error, leaving only the Hessian inverse and cross-partial derivative approximation errors to be handled separately.

**Step 4: Bounding the cross-partial derivative approximation error.** Theorem 4.8 controls the Hessian inverse approximation error. An analogous argument yields the following sample-complexity bound for the cross-partial derivative approximation error, with detailed proof in Appendix C.4.

**Proposition 4.13.** *Under Assumptions 4.1 and 4.3, for any $\delta \in (0,1)$, with probability at least $1 - \delta$, we have*

$$\|L_k^T - \nabla_{\theta,v}^2 \ell(v_k, \theta^*(v_k))\| = \mathcal{O}\left(\frac{1}{\sqrt{T}} \sqrt{1 + \log\left(\frac{1+T}{\delta}\right)}\right).$$

Finally, combining the bounds on the inner-level SGD deviation, the Hessian inverse approximation error, the cross-partial derivative approximation error and the descent inequality yields the stated convergence rate for the outer-level optimization. This completes the proof of Theorem 4.11. The full details are provided in Appendix D.2.

## 5. Numerical Experiments

In this section, we evaluate NHGD on three representative bilevel optimization tasks: (1) hyper-data cleaning (Franceschi et al., 2017), (2) data distillation (Lorraine et al., 2020), and (3) physics-informed learning for PDE-constrained optimization (Hao et al., 2023). We first demonstrate the superiority of the proposed Hessian inverse approximation over two commonly used alternatives in the standard double-loop framework: a Neumann-series approximation (**Neumann**) and a conjugate-gradient-based approximation (**CG**). We then compare NHGD with state-of-the-art baselines, including double-loop methods such as **stocBiO** (Ji et al., 2021) and **AmIGO** (Arbel & Mairal, 2022), as well as single-loop methods including **TTSA** (Hong et al., 2023) and **SOBA** (Dagréou et al., 2022). *Experimental details, including detailed problem setting and quantitative and additional results are provided in Appendix E.*

**Hyper-data Cleaning** This task learns sample weights to downweight corrupted labels (e.g., 0 for noisy samples and 1 for clean ones). We consider the MNIST dataset (LeCun et al., 1998) $\{(x_i^{\mathrm{tr}}, y_i^{\mathrm{tr}})\}_{i=1}^N$, where each label $y_i^{\mathrm{tr}} \in \{1, \ldots, C\}$ is corrupted independently with probability $p = 0.5$. The inner-level problem trains a classifier on the weighted training set, while the outer-level problem updates the weights to maximize performance on a clean validation set. The problem can be formulated as below:

$$\min_{v \in \mathbb{R}^N} \quad \frac{1}{|\mathcal{D}^{\mathrm{val}}|} \sum_{(x,y) \in \mathcal{D}^{\mathrm{val}}} l(x, y; \theta^*(v)),$$

$$\text{s.t.} \quad \theta^*(v) = \operatorname*{arg\,min}_{\theta \in \mathbb{R}^{C \times (d+1)}} \frac{1}{N} \sum_{i=1}^N \sigma(v_i) l(x_i^{\mathrm{tr}}, y_i^{\mathrm{tr}}; \theta) + \lambda \|\theta\|^2,$$

where $\lambda = 0.0001$, $l = -\log \hat{p}$, $\sigma(v) = \mathrm{Clip}(v, 0, 1)$ and the predicted probability $\hat{p}$ is

$$\hat{p}(x, y; \theta) = \frac{\exp(\langle \theta, (x; \mathbf{1}) \rangle_y)}{\sum_{j=1}^N \exp(\langle \theta, (x; \mathbf{1}) \rangle_j)}. \quad (6)$$

We then report results on a held-out test set, where successful downweighting of corrupted samples should improve generalization and test accuracy. Figure 2 compares NHGD with Neumann and CG approximations (left), and with stocBiO, AmIGO, TTSA and SOBA (right). In general, increasing the number of truncation terms improves the Neumann/CG Hessian inverse approximation. Nevertheless, NHGD converges substantially faster and achieves the highest test accuracy, even when Neumann/CG use a large number of truncation terms (e.g., 40). Moreover, compared with stocBiO, AmIGO, TTSA, and SOBA, NHGD consistently converges faster in terms of test accuracy and attains better final performance.

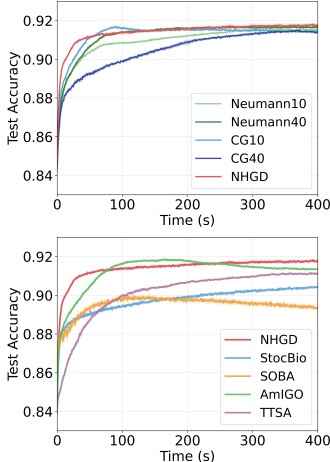

*Figure 2.* Test accuracy for the Hyper-data Cleaning task.

**Data Distillation** Data distillation aims to construct a small yet representative synthetic dataset such that models trained on it achieve performance comparable to training on the full dataset. We distill the Fashion-MNIST dataset (Xiao et al., 2017), which contains $C = 10$ classes with 6000 training images per class, into a compact synthetic set with only $n = 5$ samples per class, and formulate the task as a bilevel optimization problem: the inner problem trains a linear classifier on the distilled set, while the outer problem optimizes the synthetic samples to minimize the classifier's loss on the original training distribution. The outer variable $X \in \mathbb{R}^{d \times n \times C}$ collects all distilled samples, where $X_i^k \in \mathbb{R}^d$ denotes the $i$-th data of class $k$. The bilevel formulation is given by:

$$\min_{X \in \mathbb{R}^{d \times n \times C}} \quad \frac{1}{|\mathcal{D}^{\mathrm{val}}|} \sum_{(x,y) \in \mathcal{D}^{\mathrm{val}}} l(x, y; \theta^*(X)),$$

$$\text{s.t.} \quad \theta^*(X) = \operatorname*{arg\,min}_{\theta \in \mathbb{R}^{C \times d}} \frac{1}{nC} \sum_{k=1}^C \sum_{i=1}^n l(X_i^k, k; \theta) + \lambda \|\theta\|^2,$$

where $\lambda = 1/15680$, $l = -\log \hat{p}$ and the predicted probability is calculated by (6).

The quality of the distilled dataset is evaluated by test accuracy on a held-out test set. As shown in Figure 3 (left), NHGD converges faster and achieves higher test accuracy, even when Neumann/CG use a large number of truncation terms (e.g., 40). Moreover, as shown in Figure 3 (right), NHGD also consistently outperforms stocBiO, AmIGO, TTSA, and SOBA, converging faster in test accuracy and achieving superior final performance.

**PDE-Constrained Optimization** Bilevel optimization has recently emerged as a promising approach for PDE-constrained optimization, particularly in physics-informed machine learning (PIML). In this formulation (Hao et al., 2023), the inner-level problem solves the PDE for a given

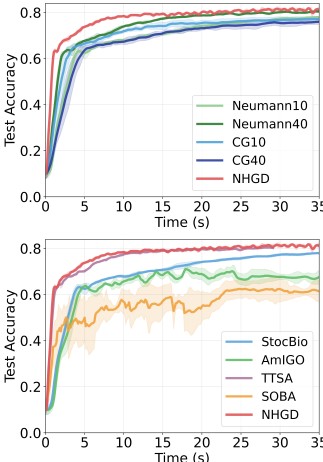

*Figure 3.* Test Accuracy for Data Distillation task.

control variable by training a state network, while the outer-level problem updates the control variable to minimize the target objective, yielding a large-scale bilevel problem with high-dimensional outer variables and a highly nonconvex inner objective. Among the baselines considered in Hao et al. (2023) (Truncated Unrolled Differentiation (Shaban et al., 2019), T1–T2 (Luketina et al., 2016), and Neumann series (Lorraine et al., 2020)), the Broyden-based method achieves the strongest performance and is therefore used as a state-of-the-art reference. We apply NHGD to the same PDE-constrained task of optimizing the time-varying temperature field of a 2D heat equation, to demonstrate its effectiveness beyond convex settings and directly compare it with the Broyden-based approach.

The original PDE-constrained optimization problem is:

$$\min_f \int_{\Omega \times [0,2]} |u - u_{\mathrm{ref}}|^2 \, dxdydt$$

$$\text{s.t.} \quad \frac{\partial u}{\partial t} - \nu \Delta u = f, \quad (x,y,t) \in \Omega \times [0,2],$$

$$u(x,y,t) = 0, \quad (x,y,t) \in \partial\Omega \times [0,2],$$

$$u(x,y,0) = 0, \quad (x,y) \in \Omega,$$

where $u(x,y,t)$ denote the state function and $f(t)$ the control function to be optimized. Following common practice in physics-informed machine learning, both the state and control are represented by neural networks: $u_\theta(x,y,t)$ and $f_v(t)$. This enables the following bilevel reformulation:

$$\min_v \int_{\Omega \times [0,2]} |u_{\theta^*(v)}(x,y,t) - u_{\mathrm{ref}}(x,y,t)|^2 \, dx \, dt,$$

$$\text{s.t.} \ \theta^*(v) = \arg\min_\theta \left[ \int_{\Omega \times [0,2]} |\mathcal{F}(u_\theta, f_v)(x,y,t)|^2 \, dx \, dt \right.$$

$$\left. + \int_{\partial\Omega \times [0,2]} |\mathcal{B}(u_\theta, f_v)(x,y,t)|^2 \, dS \, dt \right],$$

where the target state is $u_{\mathrm{ref}}(x,y,t) = 16x(1-x)y(1-y)\sin(\pi t)$, the PDE residual is $\mathcal{F}(u_\theta, f_v) := \partial u_\theta / \partial t - \nu \Delta u_\theta - f_v$ and the boundary and initial conditions are enforced through

$$\mathcal{B}(u_\theta)(x,y,t) := \begin{cases} u_\theta(x,y,t), & (x,y,t) \in \partial\Omega \times [0,2], \\ u_\theta(x,y,0), & (x,y,t) \in \Omega \times \{0\}. \end{cases}$$

Figure 4 reports the outer objective value (PDE-constrained optimization objective) versus wall-clock time, showing that NHGD achieves a comparable final objective while converging 3–4× faster than the Broyden-based method.

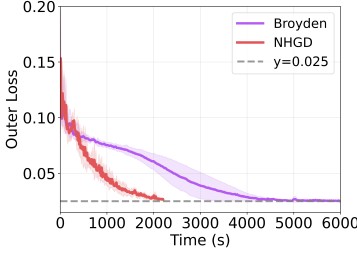

*Figure 4.* Outer loss for PDE-Constrained Optimization task

## 6. Conclusion

We propose *Natural Hypergradient Descent* (NHGD), an *optimize-and-approximate* bilevel optimization algorithm that significantly reduces computational time overhead by parallelizing hypergradient estimation with inner-level optimization. NHGD leverages the statistical equivalence between the Hessian and the FIM at the inner optimum, enabling efficient Hessian inverse approximation via stochastic gradients already computed during inner-loop SGD. We established high-probability sample complexity guarantees, showing that NHGD matches the sample complexity of existing *optimize-then-approximate* methods. Experiments on hyper–data cleaning, data distillation, and physics-informed learning for PDE-constrained optimization demonstrate that NHGD achieves comparable or superior performance to state-of-the-art baselines.

NHGD represents an initial step toward leveraging statistical structure for algorithm design in bilevel optimization. Several directions remain open for future work, including extending the statistical Hessian inverse approximation framework to single-loop algorithms and developing efficient methods for settings in which the inner-level data distribution depends on the outer variable.

## Acknowledgements

This work was supported in part by the University of Minnesota Data Science Initiative (UMN-DSI) and the NVIDIA Academic Grant Program.

## Impact Statement

This paper presents work whose goal is to advance the field of Machine Learning. There are many potential societal consequences of our work, none which we feel must be specifically highlighted here.

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

# A. Parallel Implementation of NHGD

Algorithm 1 can be efficiently parallelized across two devices. Device 1 handles the inner-loop SGD, while Device 2 synchronously approximates the terms of hypergradient. This design requires only one-directional communication (from Device 1 to Device 2) at each inner iteration, making it highly efficient for distributed implementations. The parallel implementation is detailed in Algorithm 2.

---

**Algorithm 2** Parallel Natural Hypergradient Descent

**Device 1: Inner-loop SGD**

1: **Input:** $v_1, \theta_0^T, \{\eta_t\}, T, K$
2: **for** $k = 1, \ldots, K$ **do**
3:     Initialize $\theta_k^0 = \theta_{k-1}^T$
4:     **for** $t = 0, \ldots, T-1$ **do**
5:         Sample data point $\xi_k^t$
6:         Compute gradients:
7:             $g_\theta^t = \nabla_\theta l(v_k, \theta_k^t, \xi_k^t)$
8:             $B_{\theta,v}^t = \nabla_{\theta,v}^2 l(v_k, \theta_k^t, \xi_k^t)$
9:         Update inner variable:
10:            $\theta_k^{t+1} = \Pi_\Theta \left( \theta_k^t - \eta_t g_\theta^t \right)$
11:         **Send** $g_\theta^t, B_{\theta,v}^t$ to Device 2
12:     **end for**
13:     Compute $\nabla_\theta f(v_k, \theta_k^T), \nabla_v f(v_k, \theta_k^T)$
14:     **Send** $\nabla_\theta f, \nabla_v f$ to Device 2
15:     **Receive** $\widehat{\nabla}\Phi(v_k)$ from Device 2
16:     Update outer variable:
17:         $v_{k+1} = v_k - \alpha \widehat{\nabla}\Phi(v_k)$
18: **end for**

**Device 2: Hypergradient Approximation**

1: **Input:** $A_0^T, L_0^T$
2: **for** $k = 1, \ldots, K$ **do**
3:     Initialize $A_k^0 = A_{k-1}^T, L_k^0 = L_{k-1}^T$
4:     **for** $t = 0, \ldots, T-1$ **do**
5:         **Receive** $g_\theta^t, B_{\theta,v}^t$ from Device 1
6:         Update EFIM inverse via Eq. (4):
7:         $A_k^{t+1} = \frac{t+1}{t} A_k^t - \frac{\frac{t+1}{t^2} A_k^t g_\theta^t \left( A_k^t g_\theta^t \right)^\top}{1 + \frac{1}{t} (g_\theta^t)^\top A_k^t g_\theta^t}$
8:         Update cross-partial derivative via Eq. (5):
9:         $L_k^{t+1} = \frac{t}{t+1} L_k^t + \frac{1}{t+1} B_{\theta,v}^t$
10:     **end for**
11:     **Receive** $\nabla_\theta f, \nabla_v f$ from Device 1
12:     Compute hypergradient approximation:
13:     $\widehat{\nabla}\Phi(v_k) = \nabla_v f(v_k, \theta_k^T) - (L_k^T)^\top A_k^T \nabla_\theta f(v_k, \theta_k^T)$
14:     **Send** $\widehat{\nabla}\Phi(v_k)$ to Device 1
15: **end for**

**Communication:** Yellow boxes indicate sending data; green boxes indicate receiving data. The cyan box highlights the EFIM inverse update, which is the computational bottleneck parallelized on Device 2.

---

The parallel implementation in Algorithm 2 demonstrates that NHGD naturally decomposes into two independent computational streams with minimal communication overhead. At each inner iteration, Device 1 only needs to send the computed gradients to Device 2, which independently performs the EFIM inverse and cross-partial derivative updates. This one-directional communication pattern minimizes synchronization costs and enables efficient scaling to distributed computing environments.

## B. Auxiliary Lemmas

In this section, we present auxiliary lemmas used in the proof of the main result.

**Lemma B.1.** *Assumptions 4.1, 4.3 and 4.4 imply the following:*

*(1) Gradient $\nabla_\theta l(v, \theta, \xi)$ and $\nabla_v l(v, \theta, \xi)$ are L-Lipschitz continuous with respect to $\theta$ for any $v, \xi$. $\nabla_\theta \ell(v, \theta)$ and $\nabla_v \ell(v, \theta)$ are L-Lipschitz continuous with respect to $\theta$ for any $v$.*

*(2) $\|\nabla^2_{\theta,v} l(v, \theta, \xi)\| \leq L$, for any $v, \theta, \xi$.*

*(3) $\|\nabla_\theta l(v, \theta, \xi)\| \leq LR + C_1$, for any $v, \theta, \xi$.*

*(4) Function $\ell(v, \theta)$ is $\mu$-strongly convex with respect to $\theta$ for any $v$.*

*Proof.* We first show part (1). Since function $l(v, \theta, \xi)$ is jointly $L$-smooth with respect to $(\theta, v)$, we can conclude that $\nabla_\theta l(v, \theta, \xi)$ and $\nabla_v l(v, \theta, \xi)$ are both $L$-Lipschitz continuous with respect to $\theta$ and $v$ for any $\xi$. Then, by the unbiased gradient Assumption 4.3 and Jensen's Inequality, we can obtain that $\nabla_\theta \ell(v, \theta)$ and $\nabla_v \ell(v, \theta)$ are $L$-Lipschitz continuous with respect to $\theta$ for any $v$.

Then, we show part (2). For any $x \in \mathbb{R}^{d_2}$ and unit vector $u$,

$$
\begin{aligned}
\|\nabla^2_{\theta,v} l(v, \theta, \xi) x\| &= \|\nabla_v \langle \nabla_\theta l(v, \theta, \xi), x \rangle\| \\
&= \left\| \lim_{\epsilon \to 0} \frac{\langle \nabla_\theta l(v + \epsilon u, \theta, \xi) - \nabla_\theta l(v, \theta, \xi), x \rangle}{\epsilon} \right\| \\
&= \lim_{\epsilon \to 0} \frac{\|\langle \nabla_\theta l(v + \epsilon u, \theta, \xi) - \nabla_\theta l(v, \theta, \xi), x \rangle\|}{\epsilon} \\
&\leq \lim_{\epsilon \to 0} \frac{\|\nabla_\theta l(v + \epsilon u, \theta, \xi) - \nabla_\theta l(v, \theta, \xi)\| \, \|x\|}{\epsilon} \qquad \text{(Cauchy–Schwarz Inequality)} \\
&\leq \lim_{\epsilon \to 0} \frac{L\epsilon \|u\| \, \|x\|}{\epsilon} \qquad \text{(L-Lipschitz continuity of } \nabla_\theta l \text{ in } \theta) \\
&= L \|x\|.
\end{aligned}
$$

By the definition of matrix operator norm, we have $\|\nabla^2_{\theta,v} l(v, \theta, \xi)\| = \max_{x \in \mathbb{R}^{d_1}} \dfrac{\|\nabla^2_{\theta,v} l(v, \theta, \xi) x\|}{\|x\|} \leq L$, which proves part (2).

Next, we show part (3). By gradient noise Assumption 4.4, L-smoothness of $\ell$ and the boundedness of domain $\Theta$, we obtain

$$
\begin{aligned}
\|\nabla_\theta l(v, \theta, \xi)\| &\leq \|\nabla_\theta l(v, \theta, \xi) - \nabla_\theta \ell(v, \theta)\| + \|\nabla_\theta \ell(v, \theta) - \nabla_\theta \ell(v, \theta^*(v))\| \\
&\leq C_1 + L \|\theta - \theta^*(v)\| \leq LR + C_1.
\end{aligned}
$$

Finally, we show part (4). The strongly convexity of $l(v, \theta, \xi)$ with respect to $\theta$, implies that, for any $\theta, \theta_1 \in \mathbb{R}^{d_2}$,

$$
l(v, \theta, \xi) \geq l(v, \theta_1, \xi) + \langle \nabla_\theta l(v, \theta_1, \xi), \theta - \theta_1 \rangle + \frac{\mu}{2} \|\theta - \theta_1\|^2.
$$

Taking expectation with respect to $\xi$ on both sides, and applying the unbiased stochastic gradient Assumption 4.3 yields the strong convexity of $\ell(v, \theta)$. $\qquad \square$

**Proposition B.2.** *Under Assumptions 4.1, 4.3 and 4.6, we have*

$$
I(\theta^*(v)) = H(\theta^*(v)), \qquad \forall v.
$$

*Proof.* By the definition of the Hessian of the inner objective $\ell(v, \theta)$,

$$
H\big(\theta^*(v)\big) = \nabla^2_{\theta,\theta} \ell(v, \theta^*(v)) = \nabla^2_{\theta,\theta} \mathbb{E}_{\xi \sim q(\cdot)} [l(v, \theta^*(v), \xi)] = -\nabla^2_{\theta,\theta} \mathbb{E}_{\xi \sim q(\cdot)} [\log p(\xi; v, \theta^*(v))].
$$

According to Assumption 4.3, $\nabla^2_{\theta,\theta}\ell(v,\theta) = \mathbb{E}[\nabla^2_{\theta,\theta}l(v,\theta,\xi)]$, we have

$$
\begin{aligned}
H(\theta^*(v)) &= -\mathbb{E}_{\xi\sim q(\cdot)}[\nabla^2_{\theta,\theta}\log p(\xi;v,\theta^*(v))] \\
&= -\mathbb{E}_{\xi\sim q(\cdot)}\left[\nabla_\theta \frac{\nabla_\theta p(\xi;v,\theta^*(v))}{p(\xi;v,\theta^*(v))}\right] \\
&= -\mathbb{E}_{\xi\sim q(\cdot)}\left[\frac{\nabla^2_{\theta,\theta}p(\xi;v,\theta^*(v))}{p(\xi;v,\theta^*(v))}\right] + \mathbb{E}_{\xi\sim q(\cdot)}\left[\frac{\nabla_\theta p(\xi;v,\theta^*(v))\nabla_\theta p(\xi;v,\theta^*(v))^\top}{p^2(\xi;v,\theta^*(v))}\right] \\
&= -\mathbb{E}_{\xi\sim q(\cdot)}\left[\frac{\nabla^2_{\theta,\theta}p(\xi;v,\theta^*(v))}{p(\xi;v,\theta^*(v))}\right] + \mathbb{E}_{\xi\sim q(\cdot)}\left[\nabla_\theta\left(-\log p(\xi;v,\theta^*(v))\right)\nabla_\theta\left(-\log p(\xi;v,\theta^*(v))\right)^\top\right] \\
&= -\mathbb{E}_{\xi\sim q(\cdot)}\left[\frac{\nabla^2_{\theta,\theta}p(\xi;v,\theta^*(v))}{p(\xi;v,\theta^*(v))}\right] + I(\theta^*(v)).
\end{aligned}
$$

To prove $H(\theta^*(v)) = I(\theta^*(v))$, it remains to show that the first term on the right-hand side of the previous inequality vanishes. Since $q(\cdot) = p(\cdot,v,\theta^*(v))$ (Assumption 4.6), we have

$$
\begin{aligned}
\mathbb{E}_{\xi\sim q(\cdot)}\left[\frac{\nabla^2_{\theta,\theta}p(\xi;v,\theta^*(v))}{p(\xi;v,\theta^*(v))}\right] &= \int \frac{\nabla^2_{\theta,\theta}p(\xi;v,\theta^*(v))}{p(\xi;v,\theta^*(v))}q(\xi)d\xi \\
&= \int \nabla^2_{\theta,\theta}p(\xi;v,\theta^*(v))d\xi \\
&= \nabla^2_{\theta,\theta}\int p(\xi;v,\theta^*(v))d\xi \\
&= \nabla^2_{\theta,\theta}1 \\
&= 0.
\end{aligned}
$$

Therefore, $H(\theta^*(v)) = I(\theta^*(v))$, completing the proof. $\qquad\square$

## C. Proofs for the Inner-level Analysis

In this section, we focus on the inner loop of the algorithm. We first establish the convergence rate of the inner-level SGD (Lemma 4.9). We then characterize the sample complexity of the Hessian inverse approximation (Theorem 4.8) and the cross-partial derivative approximation (Proposition 4.13).

We consider a projected SGD algorithm for minimizing the inner objective function $\ell(v,\theta) = \mathbb{E}_{\xi\sim q(\cdot)}[l(v,\theta,\xi)]$:

$$
\theta_k^{t+1} = \Pi_\Theta\left(\theta_k^t - \eta_t\nabla_\theta l(v_k,\theta_k^t,\xi_k^t)\right), \quad t = 0,1,...,T-1. \tag{7}
$$

where $\Pi_\Theta$ denotes the projection onto $\Theta = \{\theta \in \mathbb{R}^{d_2} \mid \|\theta\| \le R\}$.

We denote by $\mathcal{F}_k^{t-1} := \sigma\left(v_k,\theta_k^0,\xi_k^0,\cdots,\xi_k^{t-1}\right)$, the $\sigma$-algebra representing all information available up to iteration $(k,t-1)$. Consequently, under the conditional expectation $\mathbb{E}\left[\cdot \mid \mathcal{F}_k^{t-1}\right]$, the only remaining source of randomness is the fresh sample $\xi_k^t$ drawn at the $t$-th inner iteration.

In the subsequent analysis of the inner problem, the upper-level variable $v_k$ is fixed. For notational simplicity, we therefore omit the subscript $k$ and $v_k$. We use the simplified notation: $l(\theta_t,\xi_t) := l(v_k,\theta_k^t,\xi_k^t)$, $\ell(\theta_t) := \ell(v_k,\theta_k^t)$, $\theta^* := \theta^*(v_k)$, $\mathcal{F}_t := \mathcal{F}_k^t$, $\mathbb{E}_t[\cdot] := \mathbb{E}_{\Xi\sim q(\cdot)}\left[\cdot \mid \mathcal{F}_k^{t-1}\right]$, $I_t := I_k^t$, $A_t := A_k^t$ and $L_t := L_k^t$. We further define the stochastic gradient noise: $e_t = \nabla_\theta l(\theta_t,\xi_t) - \mathbb{E}_t[\nabla_\theta l(\theta_t,\xi_t)]$.

The algorithm for solving the inner problem is restated as follows.

---

**Algorithm 3** Solving Inner Problem with SGD

---

1: **for** $t = 0, \ldots, T-1$ **do**
2:     Sample data point $\xi_t$ from dataset and calculate $\nabla_v l(\theta_t, \xi_t)$, $\nabla_\theta l(\theta_t, \xi_t)$, and cross-partial derivative $\nabla^2_{\theta,v} l(\theta_t, \xi_t)$
3:     Update $\theta_{t+1} = \Pi_\Theta \left( \theta_t - \eta_t \nabla_\theta l(\theta_t, \xi_t) \right)$
4:     Update $A_{t+1} = \frac{t+1}{t} A_t - \frac{\frac{t+1}{t^2} A_t \nabla_\theta l(\theta_t, \xi_t)(A_t \nabla_\theta l(\theta_t, \xi_t))^\top}{1 + \frac{1}{t} \nabla_\theta l(\theta_t, \xi_t)^\top A_t \nabla_\theta l(\theta_t, \xi_t)}$
5:     Update $L_{t+1} = \frac{t}{t+1} L_t + \frac{1}{t+1} \nabla^2_{\theta,v} l(\theta_t, \xi_t)$
6: **end for**

---

## C.1. Proof of Lemma 4.9

*Proof.* By the update Eq. (7), we have

$$
\begin{aligned}
\|\theta_{t+1} - \theta^*\|^2 &= \|\Pi_\Theta(\theta_t - \eta_t \mathbb{E}_t[\nabla_\theta l(\theta_t, \xi_t)] - \eta_t e_t) - \Pi_\Theta(\theta^*)\|^2 \\
&\leq \|\theta_t - \theta^* - \eta_t \mathbb{E}_t[\nabla_\theta l(\theta_t, \xi_t)] - \eta_t e_t\|^2 \qquad (\Pi_\Theta \text{ is non-expansive w.r.t. } \|\cdot\|) \\
&= \|\theta_t - \theta^*\|^2 + \eta_t^2 \|\mathbb{E}_t[\nabla_\theta l(\theta_t, \xi_t)]\|^2 + \eta_t^2 \|e_t\|^2 \\
&\quad - 2\eta_t (\theta_t - \theta^*)^\top \mathbb{E}_t[\nabla_\theta l(\theta_t, \xi_t)] - 2\eta_t (\theta_t - \theta^*)^\top e_t + 2\eta_t^2 \mathbb{E}_t[\nabla_\theta l(\theta_t, \xi_t)]^\top e_t \\
&\leq \|\theta_t - \theta^*\|^2 + 2\eta_t^2 \|\mathbb{E}_t[\nabla_\theta l(\theta_t, \xi_t)]\|^2 + 2\eta_t^2 \|e_t\|^2 \\
&\quad - 2\eta_t (\theta_t - \theta^*)^\top \mathbb{E}_t[\nabla_\theta l(\theta_t, \xi_t)] - 2\eta_t (\theta_t - \theta^*)^\top e_t. \qquad (a^\top b \leq \tfrac{1}{2}(\|a\|^2 + \|b\|^2))
\end{aligned}
$$

Next, by the Assumption 4.3 and the $L$-smoothness of $\ell$ in $\theta$,

$$
\begin{aligned}
\|\mathbb{E}_t[\nabla_\theta l(\theta_t, \xi_t)]\|^2 &= \|\mathbb{E}_t[\nabla_\theta l(\theta_t, \xi_t) - \nabla_\theta l(\theta^*, \xi_t)]\|^2 \\
&= \|\nabla_\theta \ell(\theta_t) - \nabla_\theta \ell(\theta^*)\|^2 \\
&\leq L^2 \|\theta_t - \theta^*\|^2.
\end{aligned}
$$

The $\mu$-strong convexity of $\ell$ further implies

$$
(\theta_t - \theta^*)^\top \mathbb{E}_t[\nabla_\theta l(\theta_t, \xi_t)] = (\theta_t - \theta^*)^\top (\mathbb{E}_t[\nabla_\theta l(\theta_t, \xi_t)] - \mathbb{E}_t[\nabla_\theta l(\theta^*, \xi_t)]) \geq \mu \|\theta_t - \theta^*\|^2.
$$

Combining the preceding inequalities yields

$$
\begin{aligned}
\|\theta_{t+1} - \theta^*\|^2 &\leq (1 - 2\mu\eta_t + 2L^2\eta_t^2)\|\theta_t - \theta^*\|^2 + 2\eta_t^2 \|e_t\|^2 - 2\eta_t (\theta_t - \theta^*)^\top e_t \\
&\leq (1 - \mu\eta_t)\|\theta_t - \theta^*\|^2 + 2\eta_t^2 \|e_t\|^2 - 2\eta_t (\theta_t - \theta^*)^\top e_t. \qquad (\eta_t \leq \mu/(2L^2))
\end{aligned}
$$

For any $\lambda > 0$, since $e^{\lambda x}$ is monotone increasing in $x$, applying this to the previous inequality gives

$$
\exp(\lambda \|\theta_{t+1} - \theta^*\|^2) \leq \exp(\lambda(1 - \mu\eta_t)\|\theta_t - \theta^*\|^2) \exp(2\lambda\eta_t^2 \|e_t\|^2 - 2\lambda\eta_t (\theta_t - \theta^*)^\top e_t). \tag{8}
$$

Taking conditional expectations of (8) with respect to $\mathcal{F}_t$, and allowing the choice of $\lambda$ to depend on $t$, we obtain

$$
\begin{aligned}
&\mathbb{E}_t[\exp(\lambda_{t+1} \|\theta_{t+1} - \theta^*\|^2)] \\
&\leq \exp(\lambda_{t+1}(1 - \mu\eta_t)\|\theta_t - \theta^*\|^2) \mathbb{E}_t[\exp(2\lambda_{t+1}\eta_t^2 \|e_t\|^2 - 2\lambda_{t+1}\eta_t (\theta_t - \theta^*)^\top e_t)] \\
&\leq \exp(\lambda_{t+1}(1 - \mu\eta_t)\|\theta_t - \theta^*\|^2) \left( \mathbb{E}_t[\exp(4\lambda_{t+1}\eta_t^2 \|e_t\|^2)] \right)^{1/2} \\
&\quad \times \left( \mathbb{E}_t[\exp(-4\lambda_{t+1}\eta_t (\theta_t - \theta^*)^\top e_t)] \right)^{1/2}. \qquad \text{(conditional Cauchy–Schwarz Inequality)}
\end{aligned}
$$

Next, we bound the moment-generating function terms $\mathbb{E}_t[\exp(4\lambda_{t+1}\eta_t^2 \|e_t\|^2)]$ and $\mathbb{E}_t[\exp(-4\lambda_{t+1}\eta_t (\theta_t - \theta^*)^\top e_t)]$.

By Lemma B.1 and Jensen's Inequality, we have $\|e_t\| \leq \|\nabla_\theta l(\theta_t, \xi_t)\| + \|\mathbb{E}_t[\nabla_\theta l(\theta_t, \xi_t)]\| \leq 2(LR + C_1)$.

Thus, we have $|(\theta_t - \theta^*)^T e_t| \leq 2\|\theta_t - \theta^*\|(LR + C_1)$, which implies that $(\theta_t - \theta^*)^T e_t$ is Sub-Gaussian with variance parameter $\|\theta_t - \theta^*\|^2 (LR + C_1)^2$. By Proposition 2.5.2 in (Vershynin, 2018), there exist an absolute constant $C_a > 0$, such that

$$
\mathbb{E}_t[\exp(-4\lambda_{t+1}\eta_t (\theta_t - \theta^*)^\top e_t)] \leq \exp\left( 32 C_a^2 (LR + C_1)^2 \lambda_{t+1}^2 \eta_t^2 \|\theta_t - \theta^*\|^2 \right). \tag{9}
$$

Moreover, since $(\mathbb{E}\|e_t\|^p)^{\frac{1}{p}} \leq 2(LR + C_1)\sqrt{p}$, $\forall p \geq 1$, by Proposition 2.5.2 in (Vershynin, 2018), there exist an absolute constant $C_b > 0$, such that

$$\mathbb{E}_t[\exp(4\lambda_{t+1}\eta_t^2\|e_t\|^2)] \leq \exp(16C_b^2(LR + C_1)^2\lambda_{t+1}\eta_t^2), \tag{10}$$

with $\lambda_{t+1}$ satisfy,

$$\lambda_{t+1} \leq \frac{1}{16C_b^2(LR + C_1)^2\eta_t^2}. \tag{11}$$

Taking expectations with respect to $\mathcal{F}_t$ and applying the bounds (9) and (10), we have

$$\log\mathbb{E}[\exp(\lambda_{t+1}\|\theta_{t+1} - \theta^*\|^2)]$$

$$\leq \log\mathbb{E}\left[\exp\left(\frac{\lambda_{t+1}}{\lambda_t}\lambda_t\big(1 - \mu\eta_t + 16C_a^2(LR + C_1)^2\lambda_{t+1}\eta_t^2\big)\|\theta_t - \theta^*\|^2\right)\right.$$

$$\left. \times \exp\big(8C_b^2(LR + C_1)^2\lambda_{t+1}\eta_t^2\big)\right]$$

$$\overset{(d)}{\leq} \frac{\lambda_{t+1}}{\lambda_t}(1 - \mu\eta_t + 16C_a^2(LR + C_1)^2\lambda_{t+1}\eta_t^2)\log\mathbb{E}[\exp(\lambda_t\|\theta_t - \theta^*\|^2)]$$

$$+ 8C_b^2(LR + C_1)^2\lambda_{t+1}\eta_t^2$$

$$\overset{(e)}{\leq} \frac{\lambda_{t+1}}{\lambda_t}\left(1 - \frac{\mu\eta_t}{2}\right)\log\mathbb{E}[\exp(\lambda_t\|\theta_t - \theta^*\|^2)] + 8C_b^2(LR + C_1)^2\lambda_{t+1}\eta_t^2, \tag{12}$$

where Inequality (d) uses the choice of $\lambda_t$ ensuring

$$\frac{\lambda_{t+1}}{\lambda_t}(1 - \mu\eta_t + 16C_a^2(LR + C_1)^2\lambda_{t+1}\eta_t^2) \in (0, 1), \tag{13}$$

together with Jensen's inequality. Inequality (e) follows from choosing $\lambda_t$ such that

$$16C_a^2(LR + C_1)^2\lambda_{t+1}\eta_t \leq \frac{\mu}{2}. \tag{14}$$

Telescoping (12) over $t$ yields

$$\log\mathbb{E}[\exp(\lambda_t\|\theta_t - \theta^*\|^2)] \leq \lambda_t\prod_{j=0}^{t-1}\left(1 - \frac{\mu\eta_j}{2}\right)\|\theta_0 - \theta^*\|^2 + 8C_b^2(LR + C_1)^2\lambda_t\sum_{i=0}^{t-1}\eta_i^2\prod_{j=i+1}^{t-1}\left(1 - \frac{\mu\eta_j}{2}\right).$$

Then, applying Markov inequality, for any $\gamma > 0$, we have

$$\mathbb{P}\left(\|\theta_t - \theta^*\| \geq \gamma\right) = \mathbb{P}\left(\exp(\lambda_t\|\theta_t - \theta^*\|^2) \geq \exp(\lambda_t\gamma^2)\right)$$

$$\leq \frac{\mathbb{E}[\exp(\lambda_t\|\theta_t - \theta^*\|^2)]}{\exp(\lambda_t\gamma^2)}$$

$$\leq \exp\left(\lambda_t\prod_{j=0}^{t-1}\left(1 - \frac{\mu\eta_j}{2}\right)\|\theta_0 - \theta^*\|^2\right.$$

$$\left. + 8C_b^2(LR + C_1)^2\lambda_t\sum_{i=0}^{t-1}\eta_i^2\prod_{j=i+1}^{t-1}\left(1 - \frac{\mu\eta_j}{2}\right) - \lambda_t\gamma^2\right).$$

Equivalently, for any $\delta \in (0, 1)$, with probability at least $1 - \delta$, we have

$$\|\theta_t - \theta^*\|^2 \leq \gamma^2 = \prod_{j=0}^{t-1}\left(1 - \frac{\mu\eta_j}{2}\right)\|\theta_0 - \theta^*\|^2 + 8C_b^2(LR + C_1)^2\sum_{i=0}^{t-1}\eta_i^2\prod_{j=i+1}^{t-1}\left(1 - \frac{\mu\eta_j}{2}\right) + \frac{\log(1/\delta)}{\lambda_t}. \tag{15}$$

Substituting the diminishing stepsize $\eta_t = \frac{4}{\mu(t+q)}$ and choosing $\lambda_t = \frac{\mu}{32(LR+C_1)^2\beta\eta_{t-1}}$ (where $\beta = \max\{2C_b^2, C_a^2\}$) ensures that conditions (11), (13) and (14) are satisfied. Thus, telescoping (15) over t yields

$$\gamma^2 = \|\theta_0 - \theta^*\|^2 \prod_{i=0}^{t-1}\left(1 - \frac{2}{i+q}\right) + \frac{128}{\mu^2}C_b^2(LR+C_1)^2\sum_{i=0}^{t-1}\frac{1}{(i+q)^2}\prod_{j=i+1}^{t-1}\left(1 - \frac{2}{j+q}\right)$$

$$+ \frac{128}{\mu^2}\beta(LR+C_1)^2\frac{1}{t+q-1}\log(1/\delta)$$

$$= \|\theta_0 - \theta^*\|^2\frac{(q-2)(q-1)}{(t+q-2)(t+q-1)} + \frac{128C_b^2(LR+C_1)^2}{\mu^2(t+q-2)(t+q-1)}\sum_{i=0}^{t-1}\frac{i+q-1}{i+q}$$

$$+ \frac{128}{\mu^2}\beta(LR+C_1)^2\frac{1}{t+q-1}\log(1/\delta)$$

$$\leq \|\theta_0 - \theta^*\|^2\frac{(q-2)(q-1)}{(t+q-2)(t+q-1)} + \frac{128C_b^2(LR+C_1)^2 t}{\mu^2(t+q-1)^2} \qquad (\textstyle\sum_{i=0}^{t-1}\frac{i+q-1}{i+q} \leq t \cdot \frac{t+q-2}{t+q-1})$$

$$+ \frac{128}{\mu^2}\beta(LR+C_1)^2\frac{1}{t+q-1}\log(1/\delta)$$

$$\leq \|\theta_0 - \theta^*\|^2\left(\frac{t}{t+q}\right)^2 + \frac{256}{\mu^2}C_b^2(LR+C_1)^2\frac{1}{t+q} + \frac{256}{\mu^2}\beta(LR+C_1)^2\frac{1}{t+q}\log(1/\delta)$$

$$\stackrel{(a)}{\leq} \|\theta_0 - \theta^*\|^2\left(\frac{t}{t+q}\right)^2 + \frac{256}{\mu^2}\beta(LR+C_1)^2\left(1 + \log(1/\delta)\right)\frac{1}{t+q}. \qquad (C_b^2 \leq \beta = \max\{2C_b^2, C_a^2\})$$

Finally, we obtain that, with probability at least $1 - \delta$,

$$\|\theta_t - \theta^*\|^2 \leq \left(\frac{q}{t+q}\right)^2\|\theta_0 - \theta^*\|^2 + \frac{c'}{t+q}(1 + \log(1/\delta)),$$

and

$$\|\theta_t - \theta^*\| \leq \frac{q}{t+q}\|\theta_0 - \theta^*\| + \sqrt{\frac{c'(1+\log(1/\delta))}{t+q}},$$

where

$$c' := \frac{256}{\mu^2}(LR+C_1)^2\max\{2C_b^2, C_a^2\}. \tag{16}$$

This completes the proof. $\qquad\square$

## C.2. Statements and Proofs of Lemma C.1 and Lemma C.2

**Lemma C.1.** *Suppose Assumptions 4.1, 4.3 and 4.4 hold. For any outer iteration k, given the outer variable $v_k$, for any $\delta \in (0, 1)$, with probability at least $1 - \delta$, we have*

$$\frac{1}{T}\left\|\sum_{t=0}^{T-1}\left(\nabla_\theta l(\theta_k^t, \xi_k^t)\nabla_\theta l(\theta_k^t, \xi_k^t)^\top - \mathbb{E}_t[\nabla_\theta l(\theta_k^t, \xi_k^t)\nabla_\theta l(\theta_k^t, \xi_k^t)^\top]\right)\right\| \leq 4\sqrt{2}\,(C_1 + LR)^2\sqrt{\frac{1}{T}\log(2d_2/\delta)}.$$

*Proof.* Define the martingale increment

$$\Delta_t := \nabla_\theta l(\theta_k^t, \xi_k^t)\nabla_\theta l(\theta_k^t, \xi_k^t)^\top - \mathbb{E}_t[\nabla_\theta l(\theta_k^t, \xi_k^t)\nabla_\theta l(\theta_k^t, \xi_k^t)^\top].$$

Each $\Delta_t$ is a $d_2 \times d_2$ symmetric matrix with zero conditional mean. By the gradient boundedness from Lemma B.1 and Jensen's Inequality, we have

$$\|\Delta_t\| \leq \|\nabla_\theta l(\theta_k^t, \xi_k^t)\nabla_\theta l(\theta_k^t, \xi_k^t)^\top\| + \|\mathbb{E}_t[\nabla_\theta l(\theta_k^t, \xi_k^t)\nabla_\theta l(\theta_k^t, \xi_k^t)^\top]\|$$

$$\leq \|\nabla_\theta l(\theta_k^t, \xi_k^t)\|^2 + \mathbb{E}_t[\|\nabla_\theta l(\theta_k^t, \xi_k^t)\|^2] \leq 2(LR+C_1)^2,$$

which implies

$$\Delta_t^2 \preceq 4(LR + C_1)^4 \mathbf{I}.$$

Applying Matrix Azuma's inequality (Tropp, 2012)[Theorem 7.1], with variance parameter

$$\sigma^2 := \left\| \sum_{t=0}^{T-1} 4(LR + C_1)^4 \mathbf{I} \right\| = 4T(LR + C_1)^4,$$

we have that for any $u \geq 0$,

$$\mathbb{P}\left( \lambda_{\max}\left( \sum_{t=0}^{T-1} \Delta_t \right) \geq u \right) \leq d_2 \exp\left( -\frac{u^2}{8\sigma^2} \right). \tag{17}$$

Since $\{-\Delta_t\}_{t=0}^{T-1}$ satisfies the same conditions, we have

$$\mathbb{P}\left( \lambda_{\max}\left( -\sum_{t=0}^{T-1} \Delta_t \right) \geq u \right) \leq d_2 \exp\left( -\frac{u^2}{8\sigma^2} \right). \tag{18}$$

Using the bounds (17) and (18), we have

$$\mathbb{P}\left( \left\| \sum_{t=0}^{T-1} \Delta_t \right\| \geq u \right) = \mathbb{P}\left( \max\{ \lambda_{\max}(\sum_{t=0}^{T-1} \Delta_t), \lambda_{\max}(-\sum_{t=0}^{T-1} \Delta_t) \} \geq u \right) \qquad \text{(symmetry of } \Delta_t\text{)}$$

$$\leq \mathbb{P}\left( \lambda_{\max}\left( \sum_{t=0}^{T-1} \Delta_t \right) \geq u \right) + \mathbb{P}\left( \lambda_{\max}\left( -\sum_{t=0}^{T-1} \Delta_t \right) \geq u \right) \qquad \text{(Union Bound)}$$

$$\leq 2 d_2 \exp\left( -\frac{u^2}{8\sigma^2} \right).$$

Thus, for any $\delta \in (0, 1)$, with probability at least $1 - \delta$,

$$\frac{1}{T} \left\| \sum_{t=0}^{T-1} \Delta_t \right\| \leq 4\sqrt{2}\, (C_1 + LR)^2 \sqrt{\frac{1}{T} \log\left( 2d_2/\delta \right)}.$$

$$\square$$

**Lemma C.2.** *Suppose Assumptions 4.1, 4.3 and 4.4 hold. For any outer iteration $k$, given the outer variable $v_k$ and set stepsize as $\eta_t = \dfrac{4}{\mu\left( t + \frac{8L^2}{\mu^2} \right)}$, for any $\delta \in (0, 1)$, with probability at least $1 - \delta$, we have*

$$\frac{1}{T} \left\| \sum_{t=0}^{T-1} \left( \mathbb{E}_t[\nabla_\theta l(\theta_k^t, \xi_k^t) \nabla_\theta l(\theta_k^t, \xi_k^t)^\top] - I(\theta^*(v_k)) \right) \right\| \leq \frac{1}{\sqrt{T}} \frac{4\sqrt{2}L^2(C_1 + LR)}{\mu} \left( \frac{8L^2 R}{\mu^2} + 2\sqrt{c'(1 + \log(1/\delta))} \right).$$

*Proof.* By the definition of the FIM $I(\theta^*)$,

$$\sum_{t=0}^{T-1} \left\| \mathbb{E}_t[\nabla_\theta l(\theta_t, \xi_t) \nabla_\theta l(\theta_t, \xi_t)^\top] - I(\theta^*) \right\|$$

$$= \sum_{t=0}^{T-1} \left\| \mathbb{E}_t[\nabla_\theta l(\theta_t, \xi_t) \nabla_\theta l(\theta_t, \xi_t)^\top] - \mathbb{E}_{\xi \sim q^*(\cdot)}[\nabla_\theta l(\theta^*, \xi) \nabla_\theta l(\theta^*, \xi)^\top] \right\|$$

$$= \sum_{t=0}^{T-1} \left\| \mathbb{E}_t \left[ \nabla_\theta l(\theta_t, \xi_t) \nabla_\theta l(\theta_t, \xi_t)^\top - \nabla_\theta l(\theta^*, \xi_t) \nabla_\theta l(\theta^*, \xi_t)^\top \right] \right\|$$

$$\leq \sum_{t=0}^{T-1} \mathbb{E}_t \left[ \left\| \left( \nabla_\theta l(\theta_t, \xi_t) - \nabla_\theta l(\theta^*, \xi_t) \right) \nabla_\theta l(\theta_t, \xi_t)^\top + \nabla_\theta l(\theta^*, \xi_t) \left( \nabla_\theta l(\theta_t, \xi_t) - \nabla_\theta l(\theta^*, \xi_t) \right)^\top \right\| \right]$$

$$\leq \sum_{t=0}^{T-1} \mathbb{E}_t \left[ \|\nabla_\theta l(\theta_t, \xi_t) - \nabla_\theta l(\theta^*, \xi_t)\| \cdot \|\nabla_\theta l(\theta_t, \xi_t)\| \right] + \sum_{t=0}^{T-1} \mathbb{E}_t \left[ \|\nabla_\theta l(\theta_t, \xi_t) - \nabla_\theta l(\theta^*, \xi_t)\| \cdot \|\nabla_\theta l(\theta^*, \xi_t)\| \right]$$

$$\leq \sum_{t=0}^{T-1} \left( L\|\theta_t - \theta^*\|(LR + C_1) + L\|\theta_t - \theta^*\|(LR + C_1) \right) \qquad \text{(L-smoothness of } l \text{ with respect to } \theta)$$

$$= 2L(C_1 + LR) \sum_{t=0}^{T-1} \|\theta_t - \theta^*\|.$$

For diminishing stepsize $\eta_t = \dfrac{4}{\mu(t+q)}$, $q = \dfrac{8L^2}{\mu^2}$, by Lemma 4.9 and the projection on $\Theta$, we obtain

$$\frac{1}{T} \sum_{t=0}^{T-1} \|\theta_t - \theta^*\| \leq \frac{1}{T} \left( R \sum_{t=0}^{T-1} \frac{q}{t+q} + \sqrt{c'(1+\log(1/\delta))} \sum_{t=0}^{T-1} \frac{1}{\sqrt{t+q}} \right)$$

$$\overset{(a)}{\leq} qR \frac{\log(1 + \frac{T}{q-1})}{T} + 2\sqrt{c'(1+\log(1/\delta))} \frac{\sqrt{T+q-1}}{T}$$

$$\overset{(b)}{\leq} \frac{\sqrt{q}}{\sqrt{T}} \left( qR + 2\sqrt{c'(1+\log(1/\delta))} \right),$$

where $c'$ is an absolute constant defined in (16). Inequality (a) uses

$$\sum_{t=0}^{T-1} \frac{1}{q+t} \leq \log(1 + \frac{T}{q-1}) \quad \text{and} \quad \sum_{t=0}^{T-1} \frac{1}{\sqrt{t+q}} \leq 2\sqrt{T+q-1}.$$

Inequality (b) follows from

$$\log\left(1 + \frac{T}{q-1}\right) \leq \sqrt{T+q-1} \quad \text{and} \quad \sqrt{1 + \frac{q-1}{T}} \leq \sqrt{q}, \ \forall T \geq 1.$$

Therefore, by applying the union bound, we obtain that with probability at least $1 - T\delta$,

$$\sum_{t=0}^{T-1} \left\| \mathbb{E}_t[\nabla_\theta l(\theta_t, \xi_t) \nabla_\theta l(\theta_t, \xi_t)^\top] - I(\theta^*) \right\| \leq \frac{1}{\sqrt{T}} \frac{4\sqrt{2}L^2(C_1 + LR)}{\mu} \left( \frac{8L^2R}{\mu^2} + 2\sqrt{c'(1+\log(1/\delta))} \right).$$

$\square$

### C.3. Proof of Theorem 4.8

*Proof.* **Step 1: Reduction to EFIM error.** We aim to bound the error

$$\|A_k^T - H(\theta^*(v_k))^{-1}\| = \|(I_k^T)^{-1} - H(\theta^*(v_k))^{-1}\|,$$

where the equality is due to $I(\theta^*(v_k)) = H(\theta^*(v_k))$ (formally stated in Proposition B.2). Observe that

$$
\begin{aligned}
\|(I_k^T)^{-1} - I(\theta^*(v_k))^{-1}\| &= \|(I_k^T)^{-1}(I_k^T - I(\theta^*(v_k)))I(\theta^*(v_k))^{-1}\| \\
&\le \|(I_k^T)^{-1}\|\|I_k^T - I(\theta^*(v_k))\|\|I(\theta^*(v_k))^{-1}\| \\
&\le \frac{\|I_k^T - I(\theta^*(v_k))\|}{\sigma_{\min}(I_T)\,\mu} \qquad \text{($\mu$-strongly convex of $\ell$ by Lemma B.1)} \\
&\le \frac{\|I_k^T - I(\theta^*(v_k))\|}{\lambda_{\min}(I_T)\,\mu}.
\end{aligned} \tag{19}
$$

Since $I(\theta^*(v_k))$ and $I_k^T$ are symmetric matrix, Weyl's eigenvalue inequality yields

$$
\begin{aligned}
\lambda_{\min}(I_k^T) &\ge \lambda_{\min}(I(\theta^*(v_k))) - \lambda_{\max}(I(\theta^*(v_k)) - I_k^T) \\
&\ge \lambda_{\min}(I(\theta^*(v_k))) - \|I(\theta^*(v_k)) - I_k^T\| \\
&\ge \mu - \|I(\theta^*(v_k)) - I_k^T\|. \qquad \text{($\mu$-strongly convexity of $\ell$)}
\end{aligned}
$$

Combining the above bound with bound (19), we obtain

$$
\|(I_k^T)^{-1} - I(\theta^*(v_k))^{-1}\| \le \frac{\|I_k^T - I(\theta^*(v_k))\|}{(\mu - \|I_k^T - I(\theta^*(v_k))\|)\,\mu}.
$$

Thus, controlling the Hessian inverse approximation error reduces to bounding the error $\|I_k^T - I(\theta^*(v_k))\|$.

**Step 2: Decomposition of EFIM error.**

The main challenge of bounding $\|I_k^T - I(\theta^*(v_k))\|$ lies in the fact that

$$
I_k^T = \frac{1}{T}\sum_{t=0}^{T-1} \nabla_\theta l(\theta_k^t, \xi_k^t)\nabla_\theta l(\theta_k^t, \xi_k^t)^\top
$$

is computed along the stochastic inner-level SGD trajectory $\theta_k^0, \ldots, \theta_k^{T-1}$, whereas

$$
I(\theta^*(v_k)) = \mathbb{E}_{\xi\sim q(\cdot)}\left[\nabla_\theta l(\theta^*(v_k), \xi)\nabla_\theta l(\theta^*(v_k), \xi)^\top\right]
$$

is defined at the inner optimum. To address this mismatch, we decompose the error into a stochastic error and an optimization error as follows:

$$
\begin{aligned}
\|I_k^T - I(\theta^*(v_k))\| \le \underbrace{\frac{1}{T}\left\|\sum_{t=0}^{T-1}\left(\nabla_\theta l(\theta_k^t, \xi_k^t)\nabla_\theta l(\theta_k^t, \xi_k^t)^\top - \mathbb{E}_t[\nabla_\theta l(\theta_k^t, \xi_k^t)\nabla_\theta l(\theta_k^t, \xi_k^t)^\top]\right)\right\|}_{\text{stochastic error term } E_1} \\
+ \underbrace{\frac{1}{T}\left\|\sum_{t=0}^{T-1}\left(\mathbb{E}_t[\nabla_\theta l(\theta_k^t, \xi_k^t)\nabla_\theta l(\theta_k^t, \xi_k^t)^\top] - I(\theta^*(v_k))\right)\right\|}_{\text{optimization error term } E_2},
\end{aligned}
$$

where the inequality follows from triangular inequality.

**Step 3: Bounding stochastic and optimization error.** The stochastic error term $E_1$ is controlled using the matrix Azuma inequality. A shown in Lemma C.1, with probability at least $1 - \delta$, we have

$$
E_1 \le 4\sqrt{2}\,(C_1 + LR)^2\sqrt{\frac{1}{T}\log(2d_2/\delta)}.
$$

The optimization error term $E_2$, however, depends on how fast the inner-level SGD trajectory approaches $\theta^*(v_k)$. Under the smoothness and bounded-gradient assumptions on $l$, the bias reduces to bounding $\sum_{t=0}^{T-1}\|\theta_k^t - \theta^*(v_k)\|$. This requires a

high-probability control of the SGD iterates, by Lemma C.2, we have with probability at least $1 - T\delta$,

$$E_2 \leq \frac{4\sqrt{2}L^2(C_1 + LR)}{\mu} \frac{1}{\sqrt{T}} \left( \frac{8L^2}{\mu^2}R + 2\sqrt{c'(1 + \log(1/\delta))} \right).$$

The detailed proof of Lemmas C.1 and C.2 are provided in Appendix C.2 and C.2.

Therefore, by applying the union bound, we obtain that with probability at least $1 - (1 + T)\delta$,

$$\|I_k^T - I(\theta^*(v_k))\| \leq \frac{1}{\sqrt{T}} \left( 4\sqrt{2}(C_1 + LR)^2 \sqrt{\log(2d_2/\delta)} + \frac{4\sqrt{2}L^2(C_1 + LR)}{\mu} \left( \frac{8L^2}{\mu^2}R + 2\sqrt{c'(1 + \log(1/\delta))} \right) \right).$$

Equivalently, for any $\delta \in (0, 1)$, with probability at least $1 - \delta$, we have

$$\|I_k^T - I(\theta^*(v_k))\| \leq \frac{4\sqrt{2}(C_1 + LR)}{\sqrt{T}} \left( (C_1 + LR)\sqrt{\log\left(2d_2 \frac{1 + T}{\delta}\right)} + \frac{2L^2}{\mu}\sqrt{c'(1 + \log\left(\frac{1 + T}{\delta}\right))} + \frac{8L^4R}{\mu^3} \right).$$

**Step 4: Error aggregation and sample complexity condition.** Define

$$\gamma(t, \delta) := \frac{4\sqrt{2}(C_1 + LR)}{\sqrt{t}} \left( (C_1 + LR)\sqrt{\log\left(2d_2 \frac{1 + t}{\delta}\right)} \right)$$
$$+ \frac{4\sqrt{2}(C_1 + LR)}{\sqrt{t}} \left( \frac{2L^2}{\mu}\sqrt{c'(1 + \log\left(\frac{1 + t}{\delta}\right))} + \frac{8L^4R}{\mu^3} \right), \tag{20}$$

and

$$T_0(\delta) = \min\left\{ t \in \mathbb{N} \mid \gamma(t, \delta) \leq \frac{\mu}{2} \right\}. \tag{21}$$

Then, for any $\delta \in (0, 1)$, when $T \geq T_0(\delta)$, with probability at least $1 - \delta$, we have $\|I_k^T - I(\theta^*(v_k))\| \leq \gamma(T, \delta) \leq \frac{\mu}{2}$, and consequently,

$$\|(I_k^T)^{-1} - I(\theta^*(v_k))^{-1}\| \leq \frac{\|I_k^T - I(\theta^*(v_k))\|}{(\mu - \|I_k^T - I(\theta^*(v_k))\|)\mu} \leq \frac{2}{\mu^2}\gamma(T, \delta).$$

Finally, with probability at least $1 - \delta$, we have

$$\|A_k^T - H(\theta^*(v_k))^{-1}\| = \|(I_k^T)^{-1} - I(\theta^*(v_k))^{-1}\|$$
$$\leq \frac{\|I_k^T - I(\theta^*(v_k))\|}{(\mu - \|I_k^T - I(\theta^*(v_k))\|)\mu} = \mathcal{O}\left( \frac{1}{\sqrt{T}}\sqrt{1 + \log\left(\frac{1 + T}{\delta}\right)} \right).$$

This completes the proof. $\square$

### C.4. Statements and Proofs of Proposition 4.13 and Lemma C.3

*Proof.*

$$\left\| L_T - \nabla^2_{\theta,v}\ell(\theta^*) \right\| = \left\| \frac{1}{T}\sum_{t=0}^{T-1}\nabla^2_{\theta,v}l(\theta_t, \xi_t) - \frac{1}{T}\sum_{t=0}^{T-1}\nabla^2_{\theta,v}\ell(\theta_t) + \frac{1}{T}\sum_{t=0}^{T-1}\nabla^2_{\theta,v}\ell(\theta_t) - \nabla^2_{\theta,v}\ell(\theta^*) \right\|$$
$$\leq \frac{1}{T}\left\| \sum_{t=0}^{T-1}\left(\nabla^2_{\theta,v}l(\theta_t, \xi_t) - \nabla^2_{\theta,v}\ell(\theta_t)\right) \right\| + \frac{1}{T}\left\| \sum_{t=0}^{T-1}\left(\nabla^2_{\theta,v}\ell(\theta_t) - \nabla^2_{\theta,v}\ell(\theta^*)\right) \right\|$$
$$\overset{(a)}{\leq} \underbrace{\frac{1}{T}\left\| \sum_{t=0}^{T-1}\left(\nabla^2_{\theta,v}l(\theta_t, \xi_t) - \mathbb{E}_t[\nabla^2_{\theta,v}l(\theta_t, \xi_t)]\right) \right\|}_{E_1} + \underbrace{\frac{1}{T}\left\| \sum_{t=0}^{T-1}\left(\nabla^2_{\theta,v}\ell(\theta_t) - \nabla^2_{\theta,v}\ell(\theta^*)\right) \right\|}_{E_2},$$

where Inequality (a) follows from unbiasedness Assumption 4.3.

To bound the term $E_1$, we follow a similar argument to the proof of Theorem 4.8. We first define

$$\bar{\Delta}_t := \nabla^2_{\theta,v} l(\theta_t, \xi_t) - \mathbb{E}_t[\nabla^2_{\theta,v} l(\theta_t, \xi_t)],$$

so that by (2) of Lemma B.1, we have $\|\bar{\Delta}_t\| \leq 2L$.

Consider the self-adjoint dilation of $\bar{\Delta}_t$:

$$\mathscr{S}(\bar{\Delta}_t) := \begin{bmatrix} 0 & \bar{\Delta}_t \\ \bar{\Delta}_t^\top & 0 \end{bmatrix} \in \mathbb{R}^{(d_2+d_1)\times(d_2+d_1)}.$$

By $\lambda_{\max}(\mathscr{S}(\bar{\Delta}_t)) = \|\mathscr{S}(\bar{\Delta}_t)\| = \|\bar{\Delta}_t\|$ (Tropp, 2012)[(2.12)], we have $\lambda_{\max}(\mathscr{S}(\bar{\Delta}_t)) \leq 2L$ and hence

$$\mathscr{S}(\bar{\Delta}_t)^2 \preceq 4L^2 \mathbf{I}.$$

Applying Matrix Azuma's inequality (Tropp, 2012)[Theorem 7.1], with variance parameter

$$\sigma^2 := \|\sum_{t=0}^{T-1} 4L^2\mathbf{I}\| = 4L^2T,$$

yields, for any $u \geq 0$,

$$\mathbb{P}\left(\lambda_{\max}\left(\sum_{t=0}^{T-1} \mathscr{S}(\bar{\Delta}_t)\right) \geq u\right) \leq (d_1 + d_2)\exp\left(-\frac{u^2}{8\sigma^2}\right).$$

Repeating the symmetrization argument used in the proof of Theorem 4.8, we obtain

$$\mathbb{P}\left(\left\|\sum_{t=0}^{T-1} \mathscr{S}(\bar{\Delta}_t)\right\| \geq u\right) \leq 2(d_1 + d_2)\exp\left(-\frac{u^2}{8\sigma^2}\right).$$

Since $\|\sum_{t=0}^{T-1} \mathscr{S}(\bar{\Delta}_t)\| = \|\mathscr{S}(\sum_{t=0}^{T-1} \bar{\Delta}_t)\| = \|\sum_{t=0}^{T-1} \bar{\Delta}_t\|$, it follows that

$$\mathbb{P}\left(\left\|\sum_{t=0}^{T-1} \bar{\Delta}_t\right\| \geq u\right) \leq 2(d_1 + d_2)\exp\left(-\frac{u^2}{8\sigma^2}\right).$$

Therefore, for any $\delta \in (0, 1)$, with probability at least $1 - \delta$,

$$E_1 = \frac{1}{T}\left\|\sum_{t=0}^{T-1} \bar{\Delta}_t\right\| \leq \frac{4\sqrt{2}L}{\sqrt{T}}\sqrt{\log\left(2(d_1 + d_2)/\delta\right)}.$$

We now bound the term $E_2$. By Assumption 4.1 , $\nabla^2_{\theta,v}\ell(v, \theta)$ is Lipschitz continuous in $\theta$, therefore

$$E_2 \leq \frac{1}{T}\sum_{t=0}^{T-1}\left\|\nabla^2_{\theta,v}\ell(\theta_t) - \nabla^2_{\theta,v}\ell(\theta^*)\right\| \leq \frac{L_{l_{\theta,v}}}{T}\sum_{t=0}^{T-1}\|\theta_t - \theta^*\|.$$

Applying Lemma 4.9, with stepsize $\eta_t = \dfrac{4}{\mu(t + \frac{8L^2}{\mu^2})}$, we obtain that, for any $\delta \in (0, 1)$, with probability at least $1 - T\delta$,

$$E_2 \leq \frac{2\sqrt{2}L_{l_{\theta,v}}}{\sqrt{T}}\frac{L}{\mu}\left(\frac{8L^2R}{\mu^2} + 2\sqrt{c'(1 + \log(1/\delta))}\right),$$

where $c'$ is an absolute constant and is defined in (16).

Finally, combining the bounds for $E_1$ and $E_2$, applying the union bound, we have that with probability at least $1 - (1 + T)\delta$,

$$\left\| L_T - \nabla^2_{\theta,v}\ell(\theta^*) \right\| \leq \frac{4\sqrt{2}L}{\sqrt{T}}\sqrt{\log(2(d_1 + d_2)/\delta)} + \frac{2\sqrt{2}L_{l_{\theta,v}}}{\sqrt{T}}\frac{L}{\mu}\left(\frac{8L^2 R}{\mu^2} + 2\sqrt{c'(1 + \log(1/\delta))}\right)$$

$$= \mathcal{O}(\frac{1}{\sqrt{T}}\sqrt{1 + \log(1/\delta)}).$$

Equivalently, for any $\delta \in (0, 1)$, with probability at least $1 - \delta$, we have

$$\left\| L_T - \nabla^2_{\theta,v}\ell(\theta^*) \right\| = \mathcal{O}\left(\frac{1}{\sqrt{T}}\sqrt{1 + \log\left(\frac{1 + T}{\delta}\right)}\right).$$

This completes the proof. $\qquad\square$

**Lemma C.3.** *Under Assumptions 4.1 and 4.3, for any $\delta \in (0, 1)$, with probability at least $1 - \delta$,*

$$\|\hat{L}_k^M - \nabla^2_{\theta,v}\ell(v_k, \theta^*(v_k))\| = \mathcal{O}\left(\frac{1}{\sqrt{M}} + \sqrt{\frac{1 + \log(1/\delta)}{T}}\right).$$

*Proof.*

$$\left\| \hat{L}_M - \nabla^2_{\theta,v}\ell(\theta^*) \right\|$$

$$= \left\| \frac{1}{M}\sum_{i=1}^M \nabla^2_{\theta,v}l(\theta_T, \xi_i) - \nabla^2_{\theta,v}\ell(\theta_T) + \nabla^2_{\theta,v}\ell(\theta_T) - \nabla^2_{\theta,v}\ell(\theta^*) \right\|$$

$$\leq \frac{1}{M}\left\| \sum_{i=1}^M \left(\nabla^2_{\theta,v}l(\theta_T, \xi_i) - \nabla^2_{\theta,v}\ell(\theta_T)\right) \right\| + \left\| \nabla^2_{\theta,v}\ell(\theta_T) - \nabla^2_{\theta,v}\ell(\theta^*) \right\|$$

$$\overset{(a)}{\leq} \underbrace{\frac{1}{M}\left\| \sum_{i=1}^M \left(\nabla^2_{\theta,v}l(\theta_T, \xi_i) - \mathbb{E}_t[\nabla^2_{\theta,v}l(\theta_T, \xi_i)]\right) \right\|}_{E_1} + L_{\ell_{\theta,v}}\left\| \theta_T - \theta^* \right\|,$$

where Inequality (a) follows from the unbiasedness Assumption 4.3 and the Lipschitz continuity of $\nabla^2_{\theta,v}\ell$ from Assumption 4.1.

We now bound the term $E_1$. We first define

$$\bar{\Delta}_i := \nabla^2_{\theta,v}l(\theta_T, \xi_i) - \mathbb{E}_t[\nabla^2_{\theta,v}l(\theta_T, \xi_i)],$$

so that by (2) of Lemma B.1, we have $\|\tilde{\Delta}_i\| \leq 2L$.

Consider the self-adjoint dilation of $\tilde{\Delta}_i$:

$$\mathscr{S}(\tilde{\Delta}_i) := \begin{bmatrix} 0 & \tilde{\Delta}_i \\ \tilde{\Delta}_i^\top & 0 \end{bmatrix} \in \mathbb{R}^{(d_2 + d_1) \times (d_2 + d_1)}.$$

By $\lambda_{\max}(\mathscr{S}(\tilde{\Delta}_i)) = \|\mathscr{S}(\tilde{\Delta}_i)\| = \|\tilde{\Delta}_i\|$ (Tropp, 2012)[(2.12)], we have $\lambda_{\max}(\mathscr{S}(\bar{\Delta}_i)) \leq 2L$, and hence

$$\mathscr{S}(\bar{\Delta}_i)^2 \preceq 4L^2\mathbf{I}.$$

Applying Matrix Hoeffding inequality (Tropp, 2012)[Theorem 1.3], with variance parameter

$$\sigma^2 := \|\sum_{i=1}^M 4L^2\mathbf{I}\| = 4L^2 M,$$

yields, for any $u \geq 0$,

$$\mathbb{P}\left(\lambda_{\max}\left(\sum_{i=1}^{M}\mathscr{S}(\tilde{\Delta}_i)\right) \geq u\right) \leq (d_1 + d_2)\exp\left(-\frac{u^2}{8\sigma^2}\right).$$

Repeating the symmetrization argument used in the proof of Theorem 4.8, we obtain

$$\mathbb{P}\left(\left\|\sum_{i=1}^{M}\mathscr{S}(\tilde{\Delta}_i)\right\| \geq u\right) \leq 2(d_1 + d_2)\exp\left(-\frac{u^2}{8\sigma^2}\right).$$

Since $\|\sum_{i=1}^{M}\mathscr{S}(\tilde{\Delta}_i)\| = \|\mathscr{S}(\sum_{i=1}^{M}\tilde{\Delta}_i)\| = \|\sum_{i=1}^{M}\tilde{\Delta}_i\|$, it follows that

$$\mathbb{P}\left(\left\|\sum_{i=1}^{M}\tilde{\Delta}_i\right\| \geq u\right) \leq 2(d_1 + d_2)\exp\left(-\frac{u^2}{8\sigma^2}\right).$$

Therefore, for any $\delta \in (0,1)$, with probability at least $1 - \delta$,

$$E_1 = \frac{1}{M}\left\|\sum_{i=1}^{M}\tilde{\Delta}_i\right\| \leq \frac{4\sqrt{2}L}{\sqrt{M}}\sqrt{\log\left(2(d_1 + d_2)/\delta\right)}.$$

We now bound the term $L_{\ell_{\theta,v}}\|\theta_T - \theta^*\|$. Applying Lemma 4.9, with diminishing stepsize $\eta_t = \dfrac{4}{\mu(t + \frac{8L^2}{\mu^2})}$, we obtain that, for any $\delta \in (0,1)$, with probability at least $1 - \delta$,

$$\|\theta_T - \theta^*\| \leq \frac{\frac{8L^2}{\mu^2}}{T + \frac{8L^2}{\mu^2}}\|\theta_0 - \theta^*\| + \sqrt{\frac{c'(1 + \log(1/\delta))}{T + \frac{8L^2}{\mu^2}}},$$

where $c'$ is defined in (16).

Finally, applying the union bound, we see that with probability at least $1 - 2\delta$,

$$\left\|\hat{L}_M - \nabla^2_{\theta,v}\ell(\theta^*)\right\|$$

$$\leq \frac{4\sqrt{2}L}{\sqrt{M}}\sqrt{\log(2(d_1 + d_2)/\delta)} + \frac{\frac{8L^2R}{\mu^2}}{T + \frac{8L^2}{\mu^2}} + \sqrt{\frac{c'(1 + \log(1/\delta))}{T + \frac{8L^2}{\mu^2}}}$$

$$= \mathcal{O}(\frac{1}{\sqrt{M}} + \frac{1}{\sqrt{T}}\sqrt{1 + \log(1/\delta)}).$$

This completes the proof. □

## D. Proofs for the Outer-level Analysis

We begin by recalling two results from Lemma 2.2 of (Ghadimi & Wang, 2018), which will be used in our analysis. Under Assumptions 4.1 and 4.2, the following properties hold:

1. The mapping $\theta^*(v)$ is $\frac{L}{\mu}$-Lipschitz continuous with respect to $v$.

2. The function $\Phi(v) := f(v, \theta^*(v))$ is $L_v$-smooth with respect to $v$, where

$$L_v := \bar{L}_{f_v} + \frac{L}{\mu}\left(L_{f_v} + \bar{L}_{f_\theta}\right) + \frac{L^2}{\mu^2}L_{f\theta} + D_1\left(\frac{\bar{L}_{\ell_{\theta,v}}}{\mu} + \frac{L}{\mu^2}(\bar{L}_{\ell_{\theta,\theta}} + L_{\ell_{\theta,v}}) + \frac{L^2 L_{\ell_{\theta,\theta}}}{\mu^3}\right). \tag{22}$$

Recall that the approximate hypergradient is defined as

$$\widehat{\nabla}\Phi(v_k) := \nabla_v f(v_k, \theta_k^T) - (L_k^T)^\top A_k^T \nabla_\theta f(v_k, \theta_k^T), \tag{23}$$

and the true hypergradient is given by

$$\nabla\Phi(v_k) = \nabla_v f(v_k, \theta^*(v_k)) - \nabla_{\theta,v}^2 \ell(v_k, \theta^*(v_k))^\top H(\theta^*(v_k))^{-1} \nabla_\theta f(v_k, \theta^*(v_k)). \tag{24}$$

Accordingly, the outer-level update can be written as

$$v_{k+1} = v_k - \alpha \widehat{\nabla}\Phi(v_k).$$

## D.1. Statements and Proofs of Lemma D.1 and Lemma D.3

**Lemma D.1.** *Under Assumptions 4.1 and 4.2, and with outer stepsize $\alpha = \frac{1}{4L_v}$, we have*

$$\frac{1}{16L_v} \sum_{k=1}^K \|\nabla\Phi(v_k)\|^2 \leq \Phi(v_1) - \Phi^* + \frac{3}{16L_v} \sum_{k=1}^K \|\nabla\Phi(v_k) - \widehat{\nabla}\Phi(v_k)\|^2, \tag{25}$$

*where $L_v$ is denoted in (22) and $\Phi^*$ denotes the optimal value of the outer objective.*

*Proof.* Under Assumptions 4.1 and 4.2, $\Phi(v)$ is $L_v$-smooth with respect to $v$. Thus, we have

$$
\begin{aligned}
\Phi(v_{k+1}) &\leq \Phi(v_k) + \langle \nabla\Phi(v_k), v_{k+1} - v_k \rangle + \frac{L_v}{2}\|v_{k+1} - v_k\|^2 \\
&= \Phi(v_k) + \langle \nabla\Phi(v_k), -\alpha\widehat{\nabla}\Phi(v_k) \rangle + \frac{L_v}{2}\alpha^2\|\widehat{\nabla}\Phi(v_k)\|^2 \\
&= \Phi(v_k) + \alpha\langle \nabla\Phi(v_k), \nabla\Phi(v_k) - \widehat{\nabla}\Phi(v_k) \rangle - \alpha\|\nabla\Phi(v_k)\|^2 + \frac{L_v\alpha^2}{2}\|\widehat{\nabla}\Phi(v_k) - \nabla\Phi(v_k) + \nabla\Phi(v_k)\|^2 \\
&\overset{(a)}{\leq} \Phi(v_k) + \alpha\langle \nabla\Phi(v_k), \nabla\Phi(v_k) - \widehat{\nabla}\Phi(v_k) \rangle - (\alpha - L_v\alpha^2)\|\nabla\Phi(v_k)\|^2 + L_v\alpha^2\|\nabla\Phi(v_k) - \widehat{\nabla}\Phi(v_k)\|^2 \\
&\overset{(b)}{\leq} \Phi(v_k) + (\frac{\alpha}{2} + L_v\alpha^2)\|\nabla\Phi(v_k) - \widehat{\nabla}\Phi(v_k)\|^2 - (\frac{\alpha}{2} - L_v\alpha^2)\|\nabla\Phi(v_k)\|^2, \tag{26}
\end{aligned}
$$

where Inequality (a) follows that $\|a + b\|^2 \leq 2\|a\|^2 + 2\|b\|^2, \ \forall a, b \in \mathbb{R}^{d_1}$. Inequality (b) follows that $\langle a, b \rangle \leq \frac{1}{2}\|a\|^2 + \frac{1}{2}\|b\|^2, \ \forall a, b \in \mathbb{R}^{d_1}$.

Telescoping (26) over $k$ from 1 to $K$ and let $\alpha = \frac{1}{4L_v}$, we have

$$
\begin{aligned}
\frac{1}{16L_v} \sum_{k=1}^K \|\nabla\Phi(v_k)\|^2 &\leq \Phi(v_1) - \Phi(v_k) + \frac{3}{16L_v} \sum_{k=1}^K \|\nabla\Phi(v_k) - \widehat{\nabla}\Phi(v_k)\|^2 \\
&\leq \Phi(v_1) - \Phi^* + \frac{3}{16L_v} \sum_{k=1}^K \|\nabla\Phi(v_k) - \widehat{\nabla}\Phi(v_k)\|^2, \tag{27}
\end{aligned}
$$

where $\Phi^*$ denotes the finite optimal value of the bi-level problem. This completes the proof. $\square$

**Lemma D.2.** *Under Assumptions 4.1 and 4.2, we have that*

$$
\begin{aligned}
\|\widehat{\nabla}\Phi(v_k) - \nabla\Phi(v_k)\| &\leq \left( L_{f_v} + \frac{LL_{f_\theta}}{\mu} \right) \|\theta_k^T - \theta^*(v_k)\| + D_1 \left\| \left( L_k^T - \nabla_{\theta,v}^2 \ell(v_k, \theta^*(v_k)) \right)^\top A_k^T \right\| \\
&\quad + LD_1 \left\| A_k^T - H(\theta^*(v_k))^{-1} \right\|.
\end{aligned}
$$

*Proof.* We bound the difference of the approximate and true hypergradient as follows

$$\|\widehat{\nabla}\Phi(v_k) - \nabla\Phi(v_k)\|$$
$$= \|\nabla_v f(v_k, \theta_k^T) - (L_k^T)^\top A_k^T \nabla_\theta f(v_k, \theta_k^T) - \nabla_v f(v_k, \theta^*(v_k)) + \nabla_{\theta,v}^2 \ell(v_k, \theta^*(v_k))^\top H(\theta^*(v_k))^{-1} \nabla_\theta f(v_k, \theta^*(v_k))\|$$
$$\leq \|\nabla_v f(v_k, \theta_k^T) - \nabla_v f(v_k, \theta^*(v_k))\| + \|(L_k^T)^\top A_k^T \nabla_\theta f(v_k, \theta_k^T) - \nabla_{\theta,v}^2 \ell(v_k, \theta^*(v_k))^\top H(\theta^*(v_k))^{-1} \nabla_\theta f(v_k, \theta^*(v_k))\|$$
$$\overset{(a)}{\leq} L_{f_v} \|\theta_k^T - \theta^*(v_k)\| + \left\| \left(L_k^T - \nabla_{\theta,v}^2 \ell(v_k, \theta^*(v_k))\right)^\top A_k^T \nabla_\theta f(v_k, \theta_k^T)\right\|$$
$$+ \left\| \nabla_{\theta,v}^2 \ell(v_k, \theta^*(v_k))^\top \left(A_k^T - H(\theta^*(v_k))^{-1}\right) \nabla_\theta f(v_k, \theta_k^T)\right\|$$
$$+ \left\| \nabla_{\theta,v}^2 \ell(v_k, \theta^*(v_k)) H(\theta^*(v_k))^{-1} \left(\nabla_\theta f(v_k, \theta_k^T) - \nabla_\theta f(v_k, \theta^*(v_k))\right)\right\|$$
$$\leq L_{f_v} \|\theta_k^T - \theta^*(v_k)\| + \left\| \nabla_\theta f(v_k, \theta_k^T)\right\| \left\| \left(L_k^T - \nabla_{\theta,v}^2 \ell(v_k, \theta^*(v_k))\right)^\top A_k^T\right\|$$
$$+ \left\| \nabla_{\theta,v}^2 \ell(v_k, \theta^*(v_k))\right\| \left\| \nabla_\theta f(v_k, \theta_k^T)\right\| \left\| A_k^T - H(\theta^*(v_k))^{-1}\right\|$$
$$+ \left\| \nabla_{\theta,v}^2 \ell(v_k, \theta^*(v_k))\right\| \left\| H(\theta^*(v_k))^{-1}\right\| \left\| \nabla_\theta f(v_k, \theta_k^T) - \nabla_\theta f(v_k, \theta^*(v_k))\right\|$$
$$\overset{(b)}{\leq} L_{f_v} \|\theta_k^T - \theta^*(v_k)\| + D_1 \left\| \left(L_k^T - \nabla_{\theta,v}^2 \ell(v_k, \theta^*(v_k))\right)^\top A_k^T\right\|$$
$$+ LD_1 \left\| A_k^T - H(\theta^*(v_k))^{-1}\right\| + \frac{L}{\mu} \left\| \nabla_\theta f(v_k, \theta_k^T) - \nabla_\theta f(v_k, \theta^*(v_k))\right\|$$
$$\overset{(c)}{\leq} \left(L_{f_v} + \frac{LL_{f_\theta}}{\mu}\right) \|\theta_k^T - \theta^*(v_k)\| + D_1 \left\| \left(L_k^T - \nabla_{\theta,v}^2 \ell(v_k, \theta^*(v_k))\right)^\top A_k^T\right\|$$
$$+ LD_1 \left\| A_k^T - H(\theta^*(v_k))^{-1}\right\|,$$

where Inequality (a) follows from the Lipschitz continuity of $\nabla_v f(v, \theta)$ (Assumption 4.2) and $\|A_1 B_1 C_1 - A_2 B_2 C_2\| \leq \|(A_1 - A_2)B_1 C_1\| + \|A_2(B_1 - B_2)C_1\| + \|A_2 B_2(C_1 - C_2)\|$. Inequality (b) follows from the boundedness of $\nabla_\theta f$ (Assumption 4.2), the boundedness of $\nabla_{\theta,v}^2 \ell$ ((2) of Lemma B.1) and strongly convexity of $\ell$ ((4) of Lemma B.1). Inequality (c) follows from Lipschitz continuity of $\nabla_\theta f$ (Assumption 4.2). $\qquad\square$

**Lemma D.3.** *Under Assumptions 4.1 and 4.2, for any $\delta \in (0,1)$, with probability at least $1 - (k-1)\delta$, we have*

$$\left\|\widehat{\nabla}\Phi(v_k) - \nabla\Phi(v_k)\right\|^2 \leq \psi\Lambda_1(T)^{k-1} \left\|\theta_1^T - \theta^*(v_1)\right\|^2 + \psi b(T,\delta) \sum_{j=0}^{k-2} \Lambda_1(T)^j + \psi\Lambda_2(T) \sum_{j=1}^{k-1} \Lambda_1(T)^{k-j-1} \left\|\nabla\Phi(v_j)\right\|^2$$
$$+ \psi\Lambda_2(T) \sum_{j=1}^{k-1} \Lambda_1(T)^{k-j-1} \left(D_j^T + F_j^T\right) + 3D_k^T + 3F_k^T$$

*where $D_j^T$, $F_j^T$, $\Lambda_1(T)$), $\Lambda_2(T)$ and $b(T,\delta)$ are defined in (35), (36), (32), (33) and (34). $\psi$ is a constant defined in (39).*

*Proof.* Recall the bound provides in Lemma D.2,

$$\|\widehat{\nabla}\Phi(v_k) - \nabla\Phi(v_k)\| \leq \left(L_{f_v} + \frac{LL_{f_\theta}}{\mu}\right) \|\theta_k^T - \theta^*(v_k)\| + D_1 \left\| \left(L_k^T - \nabla_{\theta,v}^2 \ell(v_k, \theta^*(v_k))\right)^\top A_k^T\right\|$$
$$+ LD_1 \left\| A_k^T - H(\theta^*(v_k))^{-1}\right\|. \tag{28}$$

It implies that controlling the hypergradient error reduces to bounding: (i) the inner-level SGD deviation $\|\theta_k^T - \theta^*(v_k)\|$, (ii) the Hessian inverse approximation error $\|A_k^T - H(\theta^*(v_k))^{-1}\|$, and (iii) the cross-partial derivative approximation error $\|L_k^T - \nabla_{\theta,v}^2 \ell(v_k, \theta^*(v_k))\|$. We now bound the inner-loop SGD error $\|\theta_k^T - \theta^*(v_k)\|$ in this lemma. The Hessian inverse and cross-partial approximation errors, however, admit bounds independent of $v_k$, and are therefore addressed separately.

For diminishing stepsize $\eta_t = \frac{4}{\mu(t+\frac{8L^2}{\mu^2})}$, Lemma 4.9 implies that, for any $\delta \in (0,1)$, with probability of at least $1 - \delta$,

$$
\left\| \theta_k^T - \theta^*(v_k) \right\| \leq \frac{\frac{8L^2}{\mu^2}}{T + \frac{8L^2}{\mu^2}} \left\| \theta_k^0 - \theta^*(v_k) \right\| + \sqrt{\frac{c'(1+\log(1/\delta))}{T + \frac{8L^2}{\mu^2}}}
$$

$$
\overset{(d)}{\leq} \frac{\frac{8L^2}{\mu^2}}{T + \frac{8L^2}{\mu^2}} \left\| \theta_{k-1}^T - \theta^*(v_k) \right\| + \sqrt{\frac{c'(1+\log(1/\delta))}{T + \frac{8L^2}{\mu^2}}}, \tag{29}
$$

where Inequality (d) is due to the warm-start strategy $\theta_k^0 = \theta_{k-1}^T$ for the inner variable.

Furthermore, we have

$$
\begin{aligned}
\left\| \theta_{k-1}^T - \theta^*(v_k) \right\| &= \left\| \theta_{k-1}^T - \theta^*(v_{k-1}) + \theta^*(v_{k-1}) - \theta^*(v_k) \right\| \\
&\leq \left\| \theta_{k-1}^T - \theta^*(v_{k-1}) \right\| + \left\| \theta^*(v_{k-1}) - \theta^*(v_k) \right\| \\
&\leq \left\| \theta_{k-1}^T - \theta^*(v_{k-1}) \right\| + \frac{L}{\mu} \left\| v_{k-1} - v_k \right\| \qquad (\theta^*(v) \text{ is } \tfrac{L}{\mu}\text{-Lipschitz continuous}) \\
&\leq \left\| \theta_{k-1}^T - \theta^*(v_{k-1}) \right\| + \frac{\alpha L}{\mu} \left\| \widehat{\nabla}\Phi(v_{k-1}) \right\| \\
&\leq \left\| \theta_{k-1}^T - \theta^*(v_{k-1}) \right\| + \frac{\alpha L}{\mu} \left\| \widehat{\nabla}\Phi(v_{k-1}) - \nabla\Phi(v_{k-1}) \right\| + \frac{\alpha L}{\mu} \left\| \nabla\Phi(v_{k-1}) \right\|. \tag{30}
\end{aligned}
$$

Combining inequalities (28), (29) and (30), we obtain that with probability at least $1 - \delta$,

$$
\left\| \theta_k^T - \theta^*(v_k) \right\|^2 \leq \Lambda_1(T) \left\| \theta_{k-1}^T - \theta^*(v_{k-1}) \right\|^2 + \Lambda_2(T) \left( \left\| \nabla\Phi(v_{k-1}) \right\|^2 + D_{k-1}^T + F_{k-1}^T \right) + b(T,\delta), \tag{31}
$$

where

$$
\Lambda_1(T) := 5 \left( 1 + \frac{\alpha L}{\mu} \left( L_{f_v} + \frac{LL_{f_\theta}}{\mu} \right) \right)^2 \left( \frac{\frac{8L^2}{\mu^2}}{T + \frac{8L^2}{\mu^2}} \right)^2, \tag{32}
$$

$$
\Lambda_2(T) := \frac{5\alpha^2 L^2}{\mu^2} \left( \frac{\frac{8L^2}{\mu^2}}{T + \frac{8L^2}{\mu^2}} \right)^2, \tag{33}
$$

$$
b(T,\delta) := \frac{5c'(1+\log(1/\delta))}{T + \frac{8L^2}{\mu^2}}, \tag{34}
$$

$$
D_k^T := D_1^2 \left\| \left( L_k^T - \nabla_{\theta,v}^2 \ell(v_k, \theta^*(v_k)) \right)^\top A_k^T \right\|^2, \tag{35}
$$

$$
F_k^T := L^2 D_1^2 \left\| A_k^T - H(\theta^*(v_k))^{-1} \right\|^2. \tag{36}
$$

Telescoping Inequality (31) over $k$, we have that with probability at least $1 - (k-1)\delta$,

$$
\left\| \theta_k^T - \theta^*(v_k) \right\|^2 \leq \Lambda_1(T)^{k-1} \left\| \theta_1^T - \theta^*(v_1) \right\|^2 + \Lambda_2(T) \sum_{j=1}^{k-1} \Lambda_1(T)^{k-j-1} \left( \left\| \nabla\Phi(v_j) \right\|^2 + D_j^T + F_j^T \right)
$$

$$
+ b(T,\delta) \sum_{j=0}^{k-2} \Lambda_1(T)^j. \tag{37}
$$

Finally, substituting (37) into (28), we obtain the corresponding bound on the hypergradient approximation error. With probability at least $1 - (k-1)\delta$, we have

$$
\left\| \widehat{\nabla}\Phi(v_k) - \nabla\Phi(v_k) \right\|^2 \leq \psi \Lambda_1(T)^{k-1} \left\| \theta_1^T - \theta^*(v_1) \right\|^2 + \psi b(T,\delta) \sum_{j=0}^{k-2} \Lambda_1(T)^j + \psi \Lambda_2(T) \sum_{j=1}^{k-1} \Lambda_1(T)^{k-j-1} \left\| \nabla\Phi(v_j) \right\|^2
$$

$$
+ \psi \Lambda_2(T) \sum_{j=1}^{k-1} \Lambda_1(T)^{k-j-1} \left( D_j^T + F_j^T \right) + 3D_k^T + 3F_k^T, \tag{38}
$$

where

$$\psi := 3\left(L_{f_v} + \frac{LL_{f_\theta}}{\mu}\right)^2. \tag{39}$$

This completes the proof. □

### D.2. Proof of Theorem 4.11

*Proof.* **Step 1: Descent inequality for the outer level.** We begin by establishing a descent inequality for the outer-level objective. Under Assumptions 4.1 and 4.2, the overall objective is $L_v$-smooth. With the outer stepsize $\alpha = \frac{1}{4L_v}$, this can give us the following descent inequality

$$\frac{1}{16L_v}\sum_{k=1}^{K}\|\nabla\Phi(v_k)\|^2 \le \Phi(v_1) - \Phi^* + \frac{3}{16L_v}\sum_{k=1}^{K}\|\nabla\Phi(v_k) - \widehat{\nabla}\Phi(v_k)\|^2, \tag{40}$$

where $L_v$ is defined in (22) and $\Phi^*$ denotes the optimal value of the outer objective. A formal statement of this result is provided in Lemma D.1. This shows that establishing convergence guarantee reduces to bounding the hypergradient approximation error $\|\nabla\Phi(v_k) - \widehat{\nabla}\Phi(v_k)\|$.

**Step 2: Decomposition of the hypergradient approximation error.** Under the regularity conditions imposed by our assumptions, the hypergradient approximation error admits the following decomposition:

$$\|\widehat{\nabla}\Phi(v_k) - \nabla\Phi(v_k)\| \le \left(L_{f_v} + \frac{LL_{f_\theta}}{\mu}\right)\|\theta_k^T - \theta^*(v_k)\| + D_1\left\|\left(L_k^T - \nabla_{\theta,v}^2\ell(v_k, \theta^*(v_k))\right)^\top A_k^T\right\|$$
$$+ LD_1\left\|A_k^T - H(\theta^*(v_k))^{-1}\right\|,$$

where the detailed statement and proof is provided in Lemma D.2. Consequently, controlling the hypergradient error reduces to bounding: (i) the inner-level SGD deviation $\|\theta_k^T - \theta^*(v_k)\|$, (ii) the Hessian inverse approximation error $\|A_k^T - H(\theta^*(v_k))^{-1}\|$, and (iii) the cross-partial derivative approximation error $\|L_k^T - \nabla_{\theta,v}^2\ell(v_k, \theta^*(v_k))\|$.

We next bound the inner-level SGD deviation, whose error remains coupled with $\|\nabla\Phi(v_k)\|$, reflecting the intrinsic coupling between the inner SGD deviation and the outer objective. The Hessian inverse and cross-partial approximation errors, however, admit bounds independent of $v_k$, and are therefore addressed separately at the end.

**Step 3: Bounding the inner-level SGD deviation.** Lemma 4.9 shows that inner-level SGD error depends on $\|\theta_k^0 - \theta^*(v_k)\|$. To further control this, we use the standard warm-start technique $\theta_k^0 = \theta_{k-1}^T$, and the hypergradient approximation error can be further bounded as:

$$\left\|\widehat{\nabla}\Phi(v_k) - \nabla\Phi(v_k)\right\|^2 \le \psi\Lambda_1(T)^{k-1}\left\|\theta_1^T - \theta^*(v_1)\right\|^2 + \psi b(T,\delta)\sum_{j=0}^{k-2}\Lambda_1(T)^j + \psi\Lambda_2(T)\sum_{j=1}^{k-1}\Lambda_1(T)^{k-j-1}\left\|\nabla\Phi(v_j)\right\|^2$$
$$+ \psi\Lambda_2(T)\sum_{j=1}^{k-1}\Lambda_1(T)^{k-j-1}\left(D_j^T + F_j^T\right) + 3D_k^T + 3F_k^T, \tag{41}$$

where $D_j^T = D_1\|(L_j^T - \nabla_{\theta,v}^2\ell(v_j, \theta^*(v_j)))^\top A_j^T\|$, $F_j^T = LD_1\|A_j^T - (H_j^*)^{-1}\|$ and $\Lambda_1(T) = \mathcal{O}(\frac{1}{T^2})$ is defined in (32), $\Lambda_2(T) = \mathcal{O}(\frac{1}{T^2})$ is defined in (33), $b(T,\delta) = \mathcal{O}(\frac{\log(1/\delta)}{\sqrt{T}})$ is defined in (34) and $\psi$ is a constant defined in (39). The detailed statement and proof is provided in Lemma D.3.

Telescoping (41) over $k$ from 2 to $K$ and (28) (deterministic bound) for $k = 1$, applying union bound, we obtain that, with

probability at least $1 - \frac{K(K-1)}{2}\delta$,

$$
\begin{aligned}
\sum_{k=1}^{K} \left\| \nabla\Phi(v_k) - \widehat{\nabla}\Phi(v_k) \right\|^2 &\leq \psi \left\| \theta_1^T - \theta^*(v_1) \right\|^2 \sum_{k=0}^{K-1} \Lambda_1(T)^k + \psi b(T,\delta) \sum_{k=1}^{K-1}\sum_{j=0}^{k-1} \Lambda_1(T)^j \\
&\quad + \psi\Lambda_2(T) \sum_{k=2}^{K}\sum_{j=1}^{k-1} \Lambda_1(T)^{k-j-1} \left( D_j^T + F_j^T \right) + 3\sum_{k=1}^{K} \left( D_k^T + F_k^T \right) \\
&\quad + \psi\Lambda_2(T) \sum_{k=2}^{K}\sum_{j=1}^{k-1} \Lambda_1(T)^{k-j-1} \left\| \nabla\Phi(v_j) \right\|^2 .
\end{aligned}
\tag{42}
$$

For the last term in (42),

$$
\sum_{k=2}^{K}\sum_{j=1}^{k-1} \Lambda_1(T)^{k-j-1} \left\| \nabla\Phi(v_j) \right\|^2 \leq \sum_{k=0}^{K-2} \Lambda_1(T)^k \sum_{j=1}^{K-1} \left\| \nabla\Phi(v_j) \right\|^2 \leq \frac{1}{1-\Lambda_1(T)} \sum_{j=1}^{K} \left\| \nabla\Phi(v_j) \right\|^2 .
$$

Combining the above bound with (42) and descent inequality (27),

$$
\begin{aligned}
&\left( \frac{1}{16L_v} - \frac{3\psi}{16L_v} \frac{\Lambda_2(T)}{1-\Lambda_1(T)} \right) \sum_{k=1}^{K} \left\| \nabla\Phi(v_k) \right\|^2 \\
&\leq \Phi(v_1) - \Phi^* + \frac{3\psi}{16L_v} \left\| \theta_1^T - \theta^*(v_1) \right\|^2 \sum_{k=0}^{K-1} \Lambda_1(T)^k + \frac{3\psi}{16L_v} b(T,\delta) \sum_{k=1}^{K-1}\sum_{j=0}^{k-1} \Lambda_1(T)^j \\
&\quad + \frac{3\psi}{16L_v} \Lambda_2(T) \sum_{k=2}^{K}\sum_{j=1}^{k-1} \Lambda_1(T)^{k-j-1} \left( D_j^T + F_j^T \right) + \frac{9}{16L_v} \sum_{k=1}^{K} \left( D_k^T + F_k^T \right) .
\end{aligned}
\tag{43}
$$

If $T \geq \hat{T}_0 := \frac{8L^2}{\mu^2} \sqrt{ \frac{45L^2}{8L_v^2\mu^2} \left( L_{f_v} + \frac{LL_{f_\theta}}{\mu} \right)^2 + 5 \left( 1 + \frac{L}{4L_v\mu} \left( L_{f_v} + \frac{LL_{f_\theta}}{\mu} \right) \right)^2 } - \frac{8L^2}{\mu^2}$, we have the coefficient of left hand side satisfies

$$
\frac{1}{32L_v} \leq \frac{1}{16L_v} - \frac{3\psi}{16L_v} \frac{\Lambda_2(T)}{1-\Lambda_1(T)} .
$$

**Step 4: Bounding the cross-partial derivative approximation error.** Now, we derive the high-probability bound for term $D_k^T$ and $F_k^T$, which are related to the Hessian inverse approximation error and cross-partial derivative approximation error.

First, recall Proposition 4.13 implies that, exist constant $M_1 > 0$, with probability at least $1 - \delta$,

$$
\left\| L_k^T - \nabla^2_{\theta,v}\ell(v_k, \theta^*(v_k)) \right\|^2 \leq M_1 \frac{1}{T} \left( 1 + \log\left( \frac{1+T}{\delta} \right) \right), \quad \forall k, T \geq 1.
$$

Theorem 4.8 implies that, when $T \geq T_0(\delta)$ (where $T_0(\delta)$ is define at (21)), exist constant $M_2 > 0$, with probability at least $1 - \delta$,

$$
\left\| A_k^T - (H(\theta^*(v_k)))^{-1} \right\|^2 \leq M_2 \frac{1}{T} \left( 1 + \log\left( \frac{1+T}{\delta} \right) \right), \quad \forall k.
$$

Thus, applying union bound, for any $k \geq 1$, with probability at least $1 - 2\delta$,

$$
\begin{aligned}
D_k^T &= D_1^2 \left\| \left( L_k^T - \nabla^2_{\theta,v}\ell(v_k, \theta^*(v_k)) \right)^\top A_k^T \right\|^2 \\
&\leq D_1^2 \left\| L_k^T - \nabla^2_{\theta,v}\ell(v_k, \theta^*(v_k)) \right\|^2 \left\| A_k^T \right\|^2 \\
&\leq D_1^2 \left\| L_k^T - \nabla^2_{\theta,v}\ell(v_k, \theta^*(v_k)) \right\|^2 \left( 2 \left\| H(\theta^*(v_k))^{-1} \right\|^2 + 2 \left\| A_k^T - (H(\theta^*(v_k)))^{-1} \right\|^2 \right) \\
&\leq D_1^2 M_1 \frac{1}{T} \left( 1 + \log(\frac{1+T}{\delta}) \right) \left( \frac{2}{\mu^2} + 2M_2 \frac{1}{T} \left( 1 + \log(\frac{1+T}{\delta}) \right) \right) .
\end{aligned}
$$

For any $k \geq 1$, with probability at least $1 - \delta$,

$$F_k^T = L^2 D_1^2 \left\| A_k^T - (H(\theta^*(v_k)))^{-1} \right\|^2 \leq M_2 L^2 D_1^2 \frac{1}{T} \left( 1 + \log(\frac{1+T}{\delta}) \right).$$

We apply the above high-probability bounds for $D_1^T, \cdots, D_K^T, F_1^T, \cdots, F_K^T$ for bound (43). By union bound, with probability at least $1 - \frac{K(K+5)}{2}\delta$, when $T \geq T_1(\delta)$ we have

$$\frac{1}{32L_v} \frac{1}{K} \sum_{k=1}^{K} \|\nabla \Phi(v_k)\|^2$$

$$\leq \frac{\Phi(v_1) - \Phi^*}{K} + \frac{3\psi R}{16L_v} \frac{1}{K} \underbrace{\sum_{k=0}^{K-1} \Lambda_1(T)^k}_{=:E_2} + \frac{3\psi}{16L_v} \frac{1}{K} b(T, \delta) \underbrace{\sum_{k=1}^{K-1} \sum_{j=0}^{k-1} \Lambda_1(T)^j}_{=:E_3}$$

$$+ \frac{3\psi M_3}{16L_v} \frac{1}{K} \frac{1}{T} \left( 1 + \log(\frac{1+T}{\delta}) \right) \left( \frac{2+\mu^2}{\mu^2} + \frac{2M_2}{T} \left( 1 + \log(\frac{1+T}{\delta}) \right) \right) \Lambda_2(T) \underbrace{\sum_{k=2}^{K} \sum_{j=1}^{k-1} \Lambda_1(T)^{k-j-1}}_{=:E_4}$$

$$+ \frac{3M_3}{16L_v} \frac{1}{T} \left( 1 + \log(\frac{1+T}{\delta}) \right) \left( \frac{2+\mu^2}{\mu^2} + \frac{2M_2}{T} \left( 1 + \log(\frac{1+T}{\delta}) \right) \right),$$

where $M_3 = \max\{M_1 D_1^2, M_2 L^2 D_1^2\}$ and

$$T_1(\delta) = \min \left\{ t \in \mathbb{N} \mid \gamma(t, \delta) \leq \frac{\mu}{2} \text{ and } t \geq \hat{T}_0 \right\}, \tag{44}$$

and $\gamma(t, \delta)$ in defined in (20).

By the definitions of $\Lambda_1(T), \Lambda_2(T)$ and $b(T, \delta)$ in (32). (33) and (34), we have

$$E_2 = \frac{1 - \Lambda_1(T)^K}{1 - \Lambda_1(T)} \leq \frac{1}{1 - \hat{\beta} \left( \frac{q}{T+q} \right)^2},$$

$$E_3 = b(T, \delta) \sum_{k=1}^{K-1} \frac{1 - \Lambda_1(T)^k}{1 - \Lambda_1(T)} \leq b(T, \delta) K \frac{1}{1 - \Lambda_1(T)} = \frac{5c'K \left( 1 + \log(1/\delta) \right)}{T + q - \frac{\hat{\beta}q^2}{T+q}},$$

$$E_4 \leq \Lambda_2(T) K \frac{1}{1 - \Lambda_1(T)} = \frac{\hat{\varphi}q^2 K}{(T+q)^2 - \hat{\beta}q^2},$$

where

$$\hat{\beta} := 5 \left( 1 + \frac{\alpha L}{\mu} \left( L_{f_v} + \frac{LL_{f_\theta}}{\mu} \right) \right)^2 \quad \text{and} \quad \hat{\varphi} := \frac{5\alpha^2 L^2}{\mu^2}.$$

Thus, with probability at least $1 - \frac{K(K+5)}{2}\delta$, when $T \geq T_1(\delta)$, we have

$$\frac{1}{32L_v} \frac{1}{K} \sum_{k=1}^{K} \|\nabla \Phi(v_k)\|^2$$

$$\leq \frac{\Phi(v_1) - \Phi^*}{K} + \frac{3\psi R}{16L_v} \frac{1}{K} \frac{1}{1 - \hat{\beta} \left( \frac{q}{T+q} \right)^2} + \frac{15\psi c'}{16L_v} \frac{\left( 1 + \log(1/\delta) \right)}{T + q - \frac{\hat{\beta}q^2}{T+q}}$$

$$+ \frac{3\psi M_3}{16L_v} \frac{1}{T} \left( 1 + \log(\frac{1+T}{\delta}) \right) \left( \frac{2+\mu^2}{\mu^2} + \frac{2M_2}{T} \left( 1 + \log(\frac{1+T}{\delta}) \right) \right) \frac{\hat{\varphi}q^2}{(T+q)^2 - \hat{\beta}q^2}$$

$$+ \frac{3}{16L_v} \frac{1}{T} \left( 1 + \log(\frac{1+T}{\delta}) \right) \left( \frac{2+\mu^2}{\mu^2} + \frac{2M_2}{T} \left( 1 + \log(\frac{1+T}{\delta}) \right) \right).$$

Equivalently, for any $\delta \in (0,1)$, with probability at least $1 - \delta$,

$$\frac{1}{K}\sum_{k=1}^{K}\|\nabla\Phi(v_k)\|^2 \leq \frac{32L_v}{K}\left(\Phi(v_1) - \Phi^*\right) + \frac{6\psi R}{K}\frac{1}{1 - \hat{\beta}\left(\frac{q}{T+q}\right)^2} + \frac{30\psi c'}{T + q - \frac{\hat{\beta}q^2}{T+q}}\left(1 + \log\left(\frac{K(K+5)}{2\delta}\right)\right)$$

$$+ \frac{6\psi M_3(2+\mu^2)}{\mu^2 T}\left(1 + \log\left(\frac{K(K+5)(T+1)}{\delta}\right)\right)\frac{\hat{\varphi}q^2}{(T+q)^2 - \hat{\beta}q^2}$$

$$+ \frac{12\psi M_2 M_3}{T^2}\left(1 + \log\left(\frac{K(K+5)(T+1)}{\delta}\right)\right)^2\frac{\hat{\varphi}q^2}{(T+q)^2 - \hat{\beta}q^2}$$

$$+ \frac{6}{T}\left(1 + \log\left(\frac{K(K+5)(T+1)}{\delta}\right)\right)\left(\frac{2+\mu^2}{\mu^2} + \frac{2M_2}{T}\left(1 + \log\left(\frac{K(K+5)(T+1)}{\delta}\right)\right)\right)$$

$$= \mathcal{O}\left(\frac{1}{K} + \frac{\log(KT/\delta)}{T}\right).$$

This completes the proof. $\square$

# E. Details on Experiments

**Baselines.** First, we include two *controlled* baselines that share the same overall algorithmic structure as NHGD (e.g., batch size and iteration budget), differing only in the Hessian-inverse approximation strategy: **(1) CG** (Grazzi et al., 2020; Yang et al., 2023) solves the linear system $\nabla_\theta^2\ell\,x = \nabla_\theta f$ using $K$ iterations of the conjugate gradient method. **(2) Neumann** (Ji et al., 2021; Lorraine et al., 2020) estimates the inverse Hessian via a truncated Neumann series. Following standard practice, we apply a scaling parameter $\varphi$: $H^{-1}v \approx \varphi\sum_{t=0}^{T}(I - \varphi H)^t v, 0 < \varphi < 2/\lambda_{\max}(H)$.

We also compare the NHGD with several representative hypergradient-based methods. **(3) stocBiO** (Ji et al., 2021) adopts a double-loop structure. It approximates the Hessian inverse using a truncated Neumann series with $K$ iterations and an exponentially decaying batch size $B_{\text{stocBiO}}K(1 - \varphi\mu_{\text{stocBiO}})^k$ at the $k$-th Neumann step, where $\varphi$ also serves as the Neumann scaling parameter. **(4) AmIGO** (Arbel & Mairal, 2022) also follows a double-loop structure. It approximates the Hessian inverse by performing $K$ additional SGD steps to solve the quadratic subproblem $\min_x \frac{1}{2}x^\top\nabla_\theta^2\ell\,x - x^\top\nabla_\theta f$, with stepsize $\beta_{\text{amigo}}$. **(5) TTSA** (Hong et al., 2023) is a single-loop, two-time-scale method. At each epoch, it performs $K\sqrt{1+\text{epoch}}$ Neumann-series iterations with scaling parameter $\varphi$ to approximate the Hessian inverse. **(6) SOBA** (Dagréou et al., 2022) is another single-loop method that updates the inner and outer variables simultaneously.

Unless otherwise specified (e.g., the adaptive batch-size rule in stocBiO), all stochastic derivative estimates (including gradients, Hessian–vector products, and Jacobian–vector products) use a fixed batch size $D$.

**Implementation Details for NHGD.** In practice, we adopt a smoothed rank-one update for the Hessian inverse approximation, rather than the exact update in (4). Specifically, we use

$$A_k^{t+1} = \frac{1}{\beta}A_k^t - \frac{\frac{1-\beta}{\beta^2}A_k^t\nabla_\theta l(v_k,\theta_k^t,\xi_k^t)\left(A_k^t\nabla_\theta l(v_k,\theta_k^t,\xi_k^t)\right)^\top}{1 + \frac{1-\beta}{\beta}\nabla_\theta l(v_k,\theta_k^t,\xi_k^t)^\top A_k^t\nabla_\theta l(v_k,\theta_k^t,\xi_k^t)}.$$

$M$ is the batch size used to estimate the cross-partial derivative $\hat{L}_k^M = \frac{1}{M}\sum_{i=0}^{M-1}\nabla_{\theta,v}^2 l(v_k,\theta_k^T,\xi_k^i)$ as discussed in Section 3.

**Parameters Setting.** For all methods, we perform grid search over the inner stepsize $\eta$ and outer stepsize $\alpha$ in $\{0.001,\ 0.005,\ 0.01,\ 0.05,\ 0.1,\ 0.5,\ 1.0\}$. Additional hyperparameters are selected as follows: for NHGD, $\beta \in \{0.7, 0.8, 0.9, 0.95, 0.99\}$; for Neumann-series approximations, the scaling factor $\varphi \in \{0.001, 0.005, 0.01, 0.05, 0.1\}$; for AmIGO, $\beta_{\text{amigo}} \in \{0.001, 0.01, 0.1, 1.0\}$; and for stocBiO, $B_{\text{stocBiO}} \in \{5, 10, 50, 100, 200, 500\}$.

**Hardware Setup.** All experiments are conducted on NVIDIA A100-SXM4-80GB GPUs. The NHGD algorithm is implemented in a parallel two-GPU setting: the first GPU executes the inner-level SGD, while the second GPU simultaneously updates the Hessian inverse approximation.

**Reproducibility.** All reported results are averaged over 5 independent runs with random seeds $\{0, 1, 2, 3, 4\}$; we report the mean $\pm$ standard deviation across these runs. Hyperparameters selected by grid search are fixed before running the 5 seeds, and the same set of seeds is used across all baselines for fair comparison.

## E.1. Hyper-data Cleaning

**Experimental Setup.** We evaluate all methods on the MNIST dataset. The inner problem is solved using the full $50000$ training samples with $50\%$ label noise, a batch size $D = 1024$, and $10$ inner iterations. Based on the grid search, the parameters are set as: **NHGD (M=5)**: $\eta = 0.01$, $\alpha = 0.5$, $\beta = 0.8$; **stocBiO**: $\eta = 0.01$, $\alpha = 1.0$, $B_{\text{stocBiO}} = 100$, $\varphi = 0.01$, $\mu_{\text{stocBiO}} = 10^{-4}$; **AmIGO**: $\eta = 0.01$, $\alpha = 0.5$, $\beta_{\text{amigo}} = 0.001$; **TTSA**: $\eta = 0.01$, $\alpha = 0.5$, $K = 10$; **SOBA**: $\eta = 0.1$, $\alpha = 0.5$; **Neumann10 (M=5)**: $\eta = 0.005$, $\alpha = 1.0$, $\varphi = 0.1$; **Neumann40 (M=5)**: $\eta = 0.01$, $\alpha = 1.0$, $\varphi = 0.1$; **CG10 (M=5)**: $\eta = 0.01$, $\alpha = 0.5$; **CG40 (M=5)**: $\eta = 0.01$, $\alpha = 0.1$.

**Additional Results.** The outer objective (outer loss) is shown in Figure 5, and the quantitative results are shown in Table 2. The NHGD algorithm achieves the fastest convergence in terms of the outer objective and attains the highest test accuracy.

*Table 2.* Quantitative results for Data Hyper Cleaning. **Outer Loss** and **Test Accuracy** are averaged over the last 50 epochs.

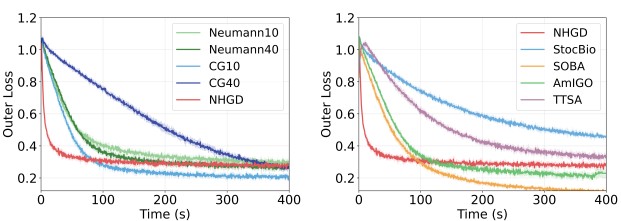

*Figure 5.* Outer loss for the Hyper-Data Cleaning task.

| Method | Outer Loss | Test Accuracy |
|---|---|---|
| stocBiO | $0.4603 \pm 0.0020$ | $0.9043 \pm 0.0004$ |
| AmIGO | $0.2117 \pm 0.0017$ | $0.9134 \pm 0.0003$ |
| TTSA | $0.2846 \pm 0.0014$ | $0.9143 \pm 0.0003$ |
| SOBA | $0.1117 \pm 0.0024$ | $0.8926 \pm 0.0011$ |
| CG10 | $0.2023 \pm 0.0017$ | $0.9141 \pm 0.0005$ |
| CG40 | $0.1827 \pm 0.0011$ | $0.9104 \pm 0.0006$ |
| Neumann10 | $0.2938 \pm 0.0020$ | $0.9153 \pm 0.0002$ |
| Neumann40 | $0.2459 \pm 0.0022$ | $0.9172 \pm 0.0003$ |
| **NHGD** | $0.2761 \pm 0.0012$ | $0.9179 \pm 0.0006$ |

## E.2. Data Distillation

**Experimental Setup.** We perform dataset distillation on the FashionMNIST dataset, which contains 10 classes and 60000 training samples. For the experiments with $n = 5$ distilled samples per class, the distilled dataset contains only 50 samples in total, so we use full-batch updates without stochastic sampling. Due to the small dataset size, stocBiO reduces to a constant–batch-size variant, making it equivalent to the Neumann baseline in this case. Based on the grid search, the parameters are set as: **NHGD (M=1)**: $\eta = 1.0$, $\alpha = 0.01$, $\beta = 0.00$; **AmIGO**: $\eta = 0.001$, $\alpha = 0.005$, $\beta_{\text{amigo}} = 1.0$; **TTSA**: $\eta = 10$, $\alpha = 0.01$, $c_{\text{in}} = 0$, $c_{\text{out}} = 0$, $K = 10$; **SOBA**: $\eta = 2.0$, $\alpha = 1.9$, $c_{\text{in}} = 0.8$, $c_{\text{out}} = 0.2$; **Neumann10 (M=1)**: $\eta = 0.1$, $\alpha = 0.005$, $\varphi = 0.1$; **Neumann40 (M=1)**: $\eta = 1.0$, $\alpha = 0.001$, $\varphi = 0.1$; **CG10 (M=1)**: $\eta = 0.001$, $\alpha = 0.001$; **CG40 (M=1)**: $\eta = 0.001$, $\alpha = 0.001$.

We also conduct the experiment with $n = 100$ distilled samples for each class. For this setting, we use batch size $D = 128$, and 30 inner iterations. Based on the grid search, the parameters are set as: **NHGD (M=5)**: $\eta = 1.0$, $\alpha = 0.5$, $\beta = 0.99$; **stocBiO**: $\eta = 1.0$, $\alpha = 1.0$, $B_{\text{stocBiO}} = 5$, $\varphi = 0.01$, $\mu_{\text{stocBiO}} = 10^{-4}$; **AmIGO**: $\eta = 0.001$, $\alpha = 1.0$, $\beta_{\text{amigo}} = 1.0$; **TTSA**: $\eta = 1.0$, $\alpha = 0.1$, $K = 10$; **SOBA**: $\eta = 0.5$, $\alpha = 1.0$; **Neumann10 (M=5)**: $\eta = 1.0$, $\alpha = 0.5$, $\varphi = 0.05$; **Neumann40 (M=5)**: $\eta = 1.0$, $\alpha = 0.1$, $\varphi = 0.1$; **CG10 (M=5)**: $\eta = 0.005$, $\alpha = 0.5$; **CG40 (M=5)**: $\eta = 0.005$, $\alpha = 0.5$.

**Additional Results.** For the experiments with $n = 5$ distilled samples per class, the distilled dataset contains only 50 samples in total, so we use full-batch updates without stochastic sampling. Due to the small dataset size, stocBiO reduces to a constant–batch-size variant, making it equivalent to the Neumann baseline in this case. Figure 6 compares the outer loss of NHGD with different baselines, while Table 3 reports the corresponding quantitative results.

We also report results for the setting with $n = 100$ distilled samples per class in Figure 7. Detailed quantitative results are summarized in Table 4. Compared with the baselines, NHGD maintains fast convergence speed (see Figure 7(a) and (c)) while achieving comparable final test accuracy (see Table 4).

## E.3. PDE-Constrained Optimization

**Experimental Setup.** The spatio-temporal domain is $\Omega = [0, 1] \times [0, 1]$ over the time interval $[0, 2]$. We sample 1024 interior collocation points and 256 boundary points on each spatial boundary ($x = 0, x = 1, y = 0, y = 1$) across the full

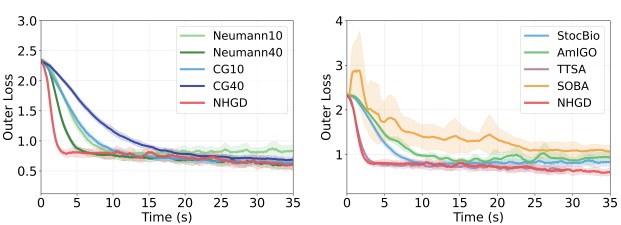

*Figure 6.* Outer Loss for Data Distillation task with $n = 5$ distilled samples per class. The outer loss curves are smoothed using a moving average with window size 10 for visualization.

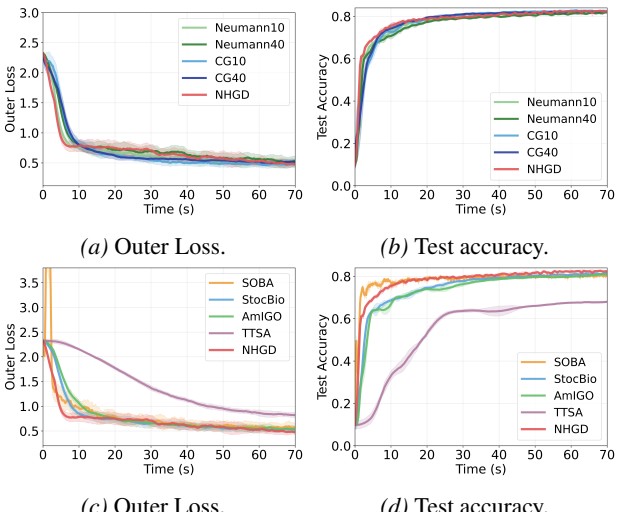

*(a)* Outer Loss.  *(b)* Test accuracy.

*(c)* Outer Loss.  *(d)* Test accuracy.

*Figure 7.* Results for the Data Distillation task with $n = 100$ distilled samples per class. The outer loss curves are smoothed using a moving average with window size 10 for visualization.

*Table 3.* Quantitative results for Data Distillation with $n = 5$ distilled samples per class. **Outer Loss** and **Test Accuracy** are the averaged over the last 50 epochs.

| Method | Outer Loss | Test Accuracy |
| --- | --- | --- |
| AmIGO | $0.8712 \pm 0.0658$ | $0.6671 \pm 0.0215$ |
| TTSA | $0.6601 \pm 0.0197$ | $0.8045 \pm 0.0019$ |
| SOBA | $0.9944 \pm 0.1043$ | $0.6468 \pm 0.0279$ |
| CG10 | $0.6289 \pm 0.0277$ | $0.7747 \pm 0.0140$ |
| CG40 | $0.6266 \pm 0.0280$ | $0.7746 \pm 0.0129$ |
| Neumann10 | $0.8587 \pm 0.0058$ | $0.7838 \pm 0.0039$ |
| Neumann40 | $0.5146 \pm 0.0125$ | $0.8202 \pm 0.0020$ |
| **NHGD** | $0.5290 \pm 0.0237$ | $0.8201 \pm 0.0048$ |

*Table 4.* Quantitative results for Data Distillation with $n = 100$ distilled samples per class. **Outer Loss** and **Test Accuracy** are averaged over last 50 epochs

| Method | Outer Loss | Test Accuracy |
| --- | --- | --- |
| StocBiO | $0.4581 \pm 0.0134$ | $0.8299 \pm 0.0015$ |
| AmIGO | $0.4930 \pm 0.0179$ | $0.8180 \pm 0.0059$ |
| TTSA | $0.7509 \pm 0.0289$ | $0.7688 \pm 0.0024$ |
| SOBA | $0.5646 \pm 0.0143$ | $0.8070 \pm 0.0031$ |
| CG10 | $0.4627 \pm 0.0178$ | $0.8245 \pm 0.0051$ |
| CG40 | $0.4982 \pm 0.0171$ | $0.8200 \pm 0.0039$ |
| Neumann10 | $0.4392 \pm 0.0119$ | $0.8316 \pm 0.0021$ |
| Neumann40 | $0.4445 \pm 0.0125$ | $0.8288 \pm 0.0043$ |
| **NHGD** | $0.4661 \pm 0.0359$ | $0.8227 \pm 0.0088$ |

time interval using Sobol quasi-random sequences. An additional 256 points are sampled at $t = 0$ to enforce the initial condition $u(x, y, 0) = 0$. For evaluating the outer-level objective, we use 512 uniformly spaced test points.

The state network $u_\theta$ is a fully connected neural network with architecture [3, 64, 64, 64, 64, 1] ( $\approx$ 12.8k parameters), which takes $(x, y, t)$ as input and outputs $u(x, y, t)$. The state network is pretrained for 2000 epochs before bilevel optimization. The control network $f_v$ has a smaller architecture [1, 32, 32, 1] and maps time $t$ to the source term $f(t)$.

**Additional Results.** We compare NHGD with the Broyden-based Hessian-inverse approximation method proposed in (Hao et al., 2023) for this challenging task. Table 5 reports the outer objective (PDE-constrained optimization loss) and PDE loss averaged over the last 50 epochs. NHGD achieves a comparable final outer loss while requiring substantially less computation time.

*Table 5.* Quantitative Performance Results for PIML. **Outer Loss** is the averaged over the last 50 epochs.

| Method | Outer Loss | PDE Loss |
| --- | --- | --- |
| Broyden | $0.0255 \pm 0.0008$ | $0.0073 \pm 0.0035$ |
| **NHGD** | $0.0266 \pm 0.0005$ | $0.0078 \pm 0.0013$ |

