# OpenReview forum: "Natural Hypergradient Descent:  Algorithm Design, Convergence Analysis, and Parallel Implementation"
_ICML.cc/2026/Conference — ICML 2026 regular_

### Official Review · Reviewer_JTJD · 2026-02-20

**Soundness:** 3
**Presentation:** 3
**Significance:** 3
**Originality:** 3
**Overall Recommendation:** 5
**Confidence:** 3

**Summary:**

The paper considers bilevel optimization probles where the outer function is smooth, and the inner problem is a smooth, and strongly convex KL minimization problem. The authors leverage this inner structure to design the Natural Hypergradient Descent algorithm. In this setting, the inner Hessian is equal to the Fisher information matrix at optimality. Using this equality, NHGD maintains an approximation of the inverse FIM and use it as a surrogate of the inverse Hessian in the hypergradient approximation. A stochastic estimator of the full-batch cross-derivative matrix is also proposed.
The algorithm is proved to achieve a sample complexity in $\mathcal{O}(\epsilon^{-2})$ with high probability. Experiments on data hyper-cleaning, data distillation and PDE constrained optimization are also provided.

**Compliance With Llm Reviewing Policy:**

Affirmed.

**Final Justification:**

The paper proposes a novel approach that is theoretically sound. Moreover, the authors appropriately addressed my concerns in their rebuttal. For these reasons, I recommend acceptance of the paper.

**Key Questions For Authors:**

* **Q1**: In the experiments, the inner problems are not purely KL minimization problems because of the $\ell^2$ regularization and, in the case of datacleaning, because of the reweighting, which is different from the setting of the paper. In those cases, is the inverse Hessian estimator still consistent or are there some adjustments? If not, how to explain the empirical behavior?

* **Q2**: A precision regarding the experiments. Since NHGD is analyzed in the setting where we have access to deterministic outer oracles, which is not the case for competitor methods, is the indicated batch size in the experimental details the inner batch size, the outer batch size, or both?

* **Q3**: In Remark 4.6, is there any condition on $\epsilon_D$ so that the analysis extends to the misspecified setting?

* **Q4**: I am not sure to understand why the cross-derivative estimate $L^t_k$ requires storing all the inner-loop graph as stated between the lines 207 and 210. Since the second line of Eq. (5) provides a recursive update, it appears possible to update $L_k^{t+1}$ on the fly by storing only $L_k^t$ and evaluating $\nabla_{v,\theta} l(v_k,\theta_k^t,\xi_k^t)$ at each inner iteration, without retaining past SGD iterates. Am I missing a dependency?

**Limitations:**

yes

**Strengths And Weaknesses:**

### Strengths
* **S1**: The paper is clearly written and easy to follow.

* **S2**: The proposed approach is novel to my knowledge

* **S3**: NHGD comes with theoretical guarantees that are sound.

* **S4**: The results are given in terms of high probability bounds, which is stronger than the bounds in expectation we usually find in the bilevel literature.


### Weaknesses

* **W1**: It would be beneficial for the motivation of the method to give some concrete examples of bilevel problems where the inner problem is exactly the minimization of a KL divergence. For instance, it seems not to be the case in the problems in the numerical experiments section due to the $\ell^2$-regularization and the reweighting in the data cleaning task.

* **W2**: In Remark 4.11, Table 1, and between lines 91 and 93 ("Overall, the NHGD matches the sample complexity of state-
of-the-art optimize-then-approximate hypergradient methods (see Table 1)"), the sample complexity of NHGD is compared to the sample complexity of other algorithms in the literature. However, the algorithms presented in Table 1 assume that the outer function is an expectation and thus use stochastic outer oracles, which is not the case of NHGD. This mismatch should be clearly stated and discussed.

* **W3**: The paper only considers deterministic oracles for the outer function whereas stochastic outer oracles are important for large-scale applications.

* **W4**: Regarding Assumption 4.5, it seems strong to me to require the distribution $p(\cdot, v, \theta^*(v))$ to match $q(\cdot)$ **for any $v$**, given that $q(\cdot)$ does not depend on $v$.

* **W5**: The code of the experiments is not provided, hindering the reproducibility.

* **W6**: I did not find the number of runs of each experiment. The compared algorithms being stochastic, there must be several runs and the plot should display the average performance over the different runs with error bars, or shaded area indicating the variability of the results across the different runs.


#### Minor

* $\Pi_\Theta$ appears for the first time in line 195, but its definition is given in line 704.

---

> ### Author Rebuttal · Authors · 2026-03-31
>
> **W1&Q1:** Thank you for this helpful question. Our algorithm is motivated by bilevel problems in which the inner objective is exactly KL-based, a structure that enables a computationally efficient and theoretically justified Fisher-based method. We also give two concrete examples that satisfy the KL setting, as shown below. To further illustrate performance in a setting that matches our theory, we also added a new large-scale learned data augmentation experiment with an 11M-parameter inner model.
>
> **(1) learned data augmentation[9].**
> - Upper: $\min_\lambda L_V(\theta^*(\lambda))$, tuning the parameters $\lambda$ of augmentation network.
> - Lower: $\theta^*(\lambda) = \arg\min\_\theta \mathbb{E}\_{p\_{\text{data}}}[-\log p\_\theta(y | f\_\lambda(x, \epsilon))]$, training a classifier on augmented data.
>
> **(2) neural architecture search[10].**
> The lower level optimizes model weights on training data, while the upper level optimizes architecture parameters on validation data. For classification, the inner objective is the standard negative log-likelihood (which is KL) and the inner training data distribution remains fixed.
>
> **W2:** As noted in the paper, since the outer-level problem is deterministic in our setting, we recompute the baseline complexities under the same deterministic-outer setting.
>
> **W3:** We agree that stochastic outer oracles are important for large-scale problems. Moving from deterministic to stochastic outer oracles mainly changes the estimation of $\nabla f(v,\theta)$, which is replaced by a sample-average estimator. We also note that our data distillation and data cleaning experiments already use stochastic outer oracles, where the method performs well empirically.
>
> We believe the theory can also be extended to this case. With stochastic outer oracles, the hypergradient estimator takes the form $\widehat{\nabla} \Phi(v_k)=\nabla_v f(v_k,\theta_k^T)+\delta_v-(L_k^T)^\top A_k^T\bigl(\nabla_\theta f(v_k,\theta_k^T)+\delta_\theta\bigr)$, where $\delta_v$, $\delta_\theta$ are the outer gradient noises. Under standard assumptions that these noises are independent of the inner stochasticity, conditionally unbiased [8,14], and satisfy boundedness conditions, the resulting outer-level error forms a bounded-difference martingale noise, so standard martingale concentration tools [12] can be used in the analysis.
>
> **W4:** We agree that a small mismatch between the learned distribution $p(\cdot;v,\theta^*(v))$ and $q(\cdot)$ may arise from model fitting error. To address this, we extend our analysis to allow for small misspecification, as shown below.
>
> Under Assumption4.1-4.4, suppose model distribution satisfies
> $\mathrm{TV}(q(\cdot), p(\cdot; v, \theta^{\ast}(v)))\leq \epsilon_D$, for all $v$. We also assume that
> $\sup_{\xi}\|\frac{\nabla\_{\theta,\theta}^2 p(\xi; v, \theta^{\ast}(v))}{p(\xi; v, \theta^{\ast}(v))}\|\leq C\_0$ for all $v$ and $2C_0 \epsilon\_D \leq\mu.$
>
> Then, for any $\delta \in (0,1)$, when $T$ is large enough, with probability at least $1-\delta$
> $\|A_T - H(\theta^{\ast})^{-1}\| \leq\mathcal{O}\left(\frac{1}{\sqrt{T}}\sqrt{1+\log\left(\frac{1+T}{\delta}\right)}+ \epsilon\_D \right)
> $ and $\frac{1}{K} \sum_{k=0}^{K-1} \| \nabla \Phi(v_k) \|^2=
> \mathcal{O}\left(\frac{1}{K}+\frac{1}{T}\log\left(\frac{KT}{\delta}\right) +\epsilon_D \right)$.
>
> **W6:** Due to the space limit, we report NHGD results averaged over 5 random seeds. We will include the averaged performance and error bar of the baselines in the revision.
> HC = data cleaning; DD-1000 = data distillation with 1000 samples, PDE=PDE-constrained problem
>
> ||HC:Outer|HC:Acc|DD-1000Outer|DD-1000Acc|PDE-Outer|PDE-pdeLoss|
> |-|-|-|-|-|-|-|
> |NHGD|0.276±0.001|0.918±0.001|0.466±0.036|0.823±0.009|0.027±0.001|0.008±0.001|
>
> **Q2:** In our experiments, the outer batching strategy varies by task in order to facilitate fair comparisons with competing methods, and all baselines use the same batching strategy within each experiment.
>
> **Q3:** Firstly, we need bound on $\frac{\nabla\_{\theta,\theta}^2 p}{p}$ is used for converting the Hessian Fisher bound into the distribution's TV distance. Secondly, the additional condition $2C\_0\epsilon\_D\le \mu$ ensures that this bias remains smaller than the curvature scale $\mu$, when bounding $\||I(\theta^{\ast})^{-1}-H(\theta^{\ast})^{-1}\||$.
>
> **Q4:** Yes, if we directly compute $L\_k^T A\_k^T \nabla\_\theta f$, then the gradient graph need not be retained. However, for efficiency,$L_k^t a$ can be compute via Jacobian-vector products instead of materializing $L\_k^t$. It first form $\sum_{i=0}^t \nabla\_{\theta}\ell(v\_k,\theta\_k^i,\xi\_k^i)a$ and then differentiate it with respect to $v$. In this case, the intermediate quantity must remain connected to $v$ through the autograd graph, so retaining the inner-loop graph may still be necessary. This creates a computational trade-off, and the best choice depends on the practical setting.
>
> Please see Reply Q7Mb for the full reference list.

---

> > ### Author Rebuttal · Reviewer_JTJD · 2026-04-01
> >
> > I thank the author for their rebuttal. My concerns have been resolved, thus I changed my score accordingly.

---

> > > ### Author Response · Authors · 2026-04-04
> > >
> > > Thank you for your constructive review comments and for updating your feedback. We greatly appreciate your time and suggestions, which have helped us improve the paper.

---

### Official Review · Reviewer_x7Yp · 2026-02-28

**Soundness:** 3
**Presentation:** 3
**Significance:** 3
**Originality:** 3
**Overall Recommendation:** 4
**Confidence:** 4

**Summary:**

This paper proposes Natural Hypergradient Descent (NHGD) for bilevel optimization. The main idea is to replace the expensive inner Hessian-inverse term in hypergradient computation with the Fisher information inverse, exploiting Hessian–Fisher equivalence at the inner optimum for KL/MLE-style inner objectives.

**Compliance With Llm Reviewing Policy:**

Affirmed.

**Final Justification:**

I appreciate the authors’ rebuttal, which has adequately addressed my main concerns. As a result, I have updated my score to 4 (Weak Accept).

**Key Questions For Authors:**

1. When the Hessian–Fisher equivalence does not hold (non-KL/MLE inner loss, misspecification, heavy-tailed noise), how does NHGD behave?
2. Please provide a simple runtime breakdown per outer step (inner SGD, inverse update, comm) and compare 1-device vs 2-device runs to substantiate the “negligible overhead” / wall-clock claims.
3. Do you maintain a dense $A_t=(I_t)^{-1}$ in all experiments? If not, what approximation (diagonal/block/K-FAC/low-rank) is used? Please report memory/time costs and how approximation quality affects convergence.

**Limitations:**

Same as the weaknesses above.

**Strengths And Weaknesses:**

Strengths:
1. The paper targets a real bottleneck in bilevel optimization: hypergradient estimation is often dominated by approximating the inner Hessian inverse / HVP, and the proposed “update the approximation during inner SGD” direction is sensible.
2. Using Fisher information as a curvature surrogate is conceptually clean for KL/MLE-style inner objectives, and the parallel implementation idea (overlapping inner optimization with curvature tracking) is practically motivated.

Weaknesses:
1. The core approximation relies on a fairly strong structural condition. It is unclear how robust the method is when the inner objective is not KL/MLE-like or when the statistical assumptions are violated; this limits generality.
2. The efficiency argument would benefit from more concrete profiling: the parallel speedup depends on hardware/communication/implementation details, but the paper does not provide a clear breakdown, like 1-device vs 2-device, compute vs comm.
3. Fisher/natural-gradient ideas and rank-one inverse updates are well-known; the paper’s contribution is mainly an integration into bilevel hypergradient estimation. Without stronger empirical evidence at scale or a clearer separation from closely related curvature-based hypergradient approximations, the impact is limited.

---

> ### Author Rebuttal · Authors · 2026-03-31
>
> **Q1&W1:** The Hessian--Fisher equivalence provides the key structure that enables us to design an efficient and provable algorithm. The behavior of NHGD beyond this setting is summarized as follows.
>
> First, outside the exact KL/MLE setting, the Fisher or empirical Fisher is still widely used as a curvature surrogate in the natural-gradient literature and performs well empirically in many non-KL ML problems [1--4]. Motivated by this, we also test NHGD on hyper-data cleaning and PDE-constrained optimization, where it remains effective.
>
> Second, a more general theory is also an important direction. In fact, we already extend our analysis to the misspecified case,  $\mathrm{TV}\left(p(\cdot; v, \theta^{\ast}(v)),q(\cdot)\right) \le \epsilon_D$, which add an $\mathcal{O}(\epsilon_D)$ term in both the Hessian-inverse approximation error and the final convergence bound.
>
> Third, heavy-tailed noise is a separate issue from the Hessian--Fisher equivalence, as it concerns the stochastic inner-gradient noise. In our algorithm, the projection step (Line 5, Eq. (3)) keeps the iterates in a bounded feasible set. If the stochastic gradient estimator is Lipschitz continuous, then the stochastic gradients and noise remain uniformly bounded, providing a basic stability mechanism. We agree that handling genuinely heavy-tailed gradient noise is an important direction for future work, but it is beyond the scope of the current paper.
>
> **Q2&W2:** The table below reports per-epoch runtime on 1 and 2 devices, with a component-wise breakdown for the 2-GPU implementation. It shows that inner-SGD gradients can be reused for Hessian-inverse updates with essentially no communication overhead, since gradient transfer overlaps with inner-SGD computation. All times are in ms.
>
> ||1-GPU (Sequential)|2-GPU (Parallel)|GPU0(Inner SGD)|GPU0(Comm. to 1)|GPU1($A^t$ update)|GPU1(Comm. to 0)|
> |-|-|-|-|-|-|-|
> |Data Hyper-Cleaning|$44.2 \pm 0.6$|$34.8 \pm 0.2$|$17.1 \pm 0.1$|$0.73 \pm 0.02$|$13.5 \pm 0.01$|$0.06 \pm 0.01$|
> |Data Distillation (1000 distilled samples)|$85.1 \pm 1.0$| $49.2 \pm 5.4$|$40.1 \pm 2.0$|$2.04 \pm 0.2$|$40.4 \pm 0.1$|$0.05 \pm 0.01$|
> |PDE-constrained Opt|$6486.5 \pm 135.0$|$6006.6 \pm 233.2$|$1323.7 \pm 76.2$|$8.45 \pm 0.28$|$419.2 \pm 0.5$|$0.06 \pm 0.00$|
>
> **W3:** Our contribution is not simply combining Fisher/natural-gradient ideas with bilevel optimization. The key distinction is that our method is derived from the hypergradient computation itself: we approximate the inverse Hessian required by the implicit function theorem, rather than using the Fisher matrix to precondition parameter updates as in natural-gradient methods.
>
> Under Assumption 4.5, hypergradient computation requires the inverse Hessian. The Hessian--Fisher equivalence, $H^{\ast} = \mathbb{E}\_{\xi\sim q}[\nabla\_\theta \log p(\xi,v,\theta^{\ast}(v)) \nabla\_\theta \log p(\xi,v,\theta^{\ast}(v))^\top]$, directly motivates our running-average estimator of gradient outer products, which provably converges to $H^*$ and happens to have the same form as the EFIM.
>
> This is fundamentally different from natural-gradient methods, which use the Fisher matrix under the model distribution, $F(\theta)=\mathbb{E}\_{\xi \sim p_\theta}[\nabla\_\theta \log p(\xi;\theta)\nabla\_\theta \log p(\xi;\theta)^\top]$. Although some works in the natural-gradient literature replace this with the EFIM, leading to an update that looks similar to ours, the EFIM is merely a computational surrogate, lacks the same theoretical justification [11].
>
> We also agree that stronger large-scale evidence is valuable. Beyond the PDE experiment, we have added a learned data augmentation experiment on CIFAR-10 with ResNet-18; see our reply to Q3.
>
> **Q3:** For small-scale problems, we maintain a dense $A_t$.
>
> For large problems, this is infeasible: ResNet-18 has 11.2M parameters, so the full Hessian is about 470TB. To address this, we use KFAC in the CIFAR-10 learned data augmentation experiment with a ResNet-18 inner model, following [9]. KFAC factorizes provides the layerwise approximation and reduces the computation to small per-layer inversions (at most $576 \times 576$).
>
> For ResNet-18, the KFAC factors and cached inverses require 3.35GB on a second GPU, versus 2.28GB for training on the first GPU. All KFAC computations run on the second GPU in parallel with inner-loop training on the first GPU and are fully overlapped: communication overhead is 0.0s, and the first GPU waits less than 1s over 200 epochs for the precomputed inverses.
>
> |Component|GPU0 (training)|GPU1 (KFAC)|
> |-|-|-|
> |Inner-loop time|1465s|—|
> |KFAC EMA + inversions|—|(overlapped)|
> |Hypergradient apply|—| 436s (cum.)|
> |Total time|4948±10s||
>
> NHGD with KFAC achieves comparable test accuracy to the Neumann baseline [9], while reducing wall-clock time from $5985$s to $4948$ s.
>
> |Method|Mean test acc|Mean time|
> |-|-|-|
> |Neumann[9]|0.9381±0.002|5985±47s|
> |NHGD|0.9384±0.001|4948±10s|
>
> Please see Reply Q7Mb for the full reference list.

---

> > ### Author Rebuttal · Reviewer_x7Yp · 2026-03-31
> >
> > The rebuttal has addressed my concerns, and I will update my score to 4 (Weak Accept).

---

> > > ### Author Response · Authors · 2026-04-04
> > >
> > > Thank you for your constructive review comments and for updating your feedback. We greatly appreciate your time and suggestions, which have helped us improve the paper.

---

### Official Review · Reviewer_JyLm · 2026-03-08

**Soundness:** 2
**Presentation:** 3
**Significance:** 3
**Originality:** 3
**Overall Recommendation:** 4
**Confidence:** 3

**Summary:**

The paper proposes Natural Hypergradient Descent (NHGD) to address the bottleneck of hypergradient estimation in bilevel optimization. By assuming the inner problem is a KL divergence minimization, it uses the empirical Fisher information matrix (EFIM) inverse as a surrogate for the inner Hessian inverse. This allows for a parallel 'optimize-and-approximate' framework with an $\tilde{\mathcal{O}}(\epsilon^{-2})$ sample complexity. Experiments show speedups over Neumann/CG baselines on hyper-data cleaning, distillation, and a PDE task.

**Compliance With Llm Reviewing Policy:**

Affirmed.

**Final Justification:**

I thank the authors for their detailed rebuttal and the additional experiments. The new results adequately address my concerns regarding the algorithm's scalability on more complex architectures. While my theoretical reservations about Assumption 4.5 remain a limitation in practice, the empirical evidence is now strong enough to support the paper's core claims. I am maintaining my current positive score of Weak Accept.

**Key Questions For Authors:**

1 Given that Assumption 4.5 is practically impossible to satisfy in constrained bilevel setups, why use the EFIM instead of the True FIM? Standard modern natural gradient methods (like K-FAC) sample from the predictive distribution precisely to avoid this well-specified assumption. Would the framework and guarantees hold if True FIM were used?

2 Can you provide results on more standard, complex architectures (e.g., ResNets) or datasets (e.g., CIFAR-10) to support the claims of large-scale efficiency?

**Limitations:**

Yes

**Strengths And Weaknesses:**

Strengths

1 Effectively translates natural gradient theory into the bilevel setting, providing a creative, parallelizable alternative to standard sequential Neumann/CG approximations.

2 Achieves a highly competitive $\tilde{\mathcal{O}}(\epsilon^{-2})$ sample complexity and provides strict high-probability error bounds, which are stronger than typical expectation-based bounds in this literature.

3 The empirical evaluation tests against a solid mix of low-level numerical approximations (Neumann, CG) and modern single/double-loop bilevel frameworks (TTSA, SOBA, stocBiO, AmIGO).

Weaknesses

1 The theoretical guarantees (Theorems 4.7 and 4.10) require the inner minimizer to perfectly match the true data distribution ($p(\cdot; v, \theta^*(v)) = q(\cdot)$). This "well-specified" assumption is highly unrealistic in a bilevel setting, where the inner problem is heavily constrained/distorted by the outer variable (e.g., in data cleaning/distillation).

2 The entire FIM-Hessian equivalence relies on the inner objective being a KL divergence. Yet, the PDE-constrained optimization task uses Mean Squared Error (MSE) to fit residuals. There is no theoretical justification for why the EFIM approximation remains valid here.

3 Claims of "large-scale" applicability are not supported by the experiments, which are restricted to MNIST/FashionMNIST and small MLPs/linear classifiers.

---

> ### Author Rebuttal · Authors · 2026-03-30
>
> **W1:** Thank you for this comment. Our algorithm is motivated by applications such as neural architecture search [10] and learned data augmentation [9], where this assumption is reasonable.
>
> More generally, we agree that exact well-specifiedness may not hold in many bilevel settings. To address this, our analysis also allows for distribution misspecification： $\mathrm{TV}\left(p(\cdot; v, \theta^*(v)),q(\cdot)\right) \le \epsilon_D$. In this case, misspecification contributes only an additional $\mathcal{O}(\epsilon_D)$ term to the Hessian-inverse approximation error and the final convergence bound.
>
> We also note that prior hypergradient-based bilevel analyses [6-8] typically assume a fixed inner data distribution independent of the outer variable. However, allowing the inner distribution to depend on the outer variable is indeed important, and we view this as an interesting direction for future work.
>
> **W2:** We use the PDE-constrained problem to demonstrate the algorithm’s empirical performance in a non-KL setting. Using Fisher-based curvature approximations outside the exact KL/Fisher--Hessian equivalence regime is common in practice and consistent with the natural-gradient literature [3,4]. Such methods have shown strong empirical performance even when exact equivalence does not hold. In particular, recent work in physics-informed machine learning shows that natural-gradient methods can be effective for PINN training despite the non-KL objective [1,2]. We include this PDE-constrained optimization task to illustrate that NHGD can also perform well beyond the KL-divergence setting.
>
> **Q1:** Thank you for this question, which helps clarify the difference between natural gradient descent and NHGD. Our method is motivated by applications such as neural architecture search [10] and learned data augmentation [9], where Assumption 4.5 holds. We exploit this structure to design an efficient hypergradient approximation for bilevel optimization.
>
> Our approximation is derived from the implicit function theorem (line 32-35), which requires estimating the Hessian inverse $(H^{*})^{-1}=(\nabla_{\theta,\theta}^2\mathbb{E}\_{\xi\sim q}[-\log p(\xi,v,\theta^{\ast}(v))])^{-1}$ appearing in the hypergradient. Under the Hessian--Fisher equivalence assumption, the Hessian satisfies $H^{\ast}=I^{\ast}=\mathbb{E}\_{\xi\sim q}[\nabla\_\theta \log p(\xi,v,\theta^{\ast}(v))\nabla\_\theta \log p(\xi,v,\theta^{\ast}(v))^\top]$.  This motivates using a running average of gradient outer products, which provably converges to $H^{\ast}$. This estimator happens to have the same form as the EFIM.
>
> Importantly, the expectation here is taken with respect to the data distribution $q$, as required by the hypergradient derivation. This differs from natural-gradient methods such as K-FAC, where the Fisher matrix is taken under the model distribution, $F(\theta)=\mathbb{E}\_{\xi \sim p\_\theta}[\nabla\_\theta \log p(\xi;\theta)\nabla\_\theta \log p(\xi;\theta)^\top]$, and is used to capture the local geometry of the parameterization. In a bilevel setting, replacing the above quantity with the FIM would in general introduce bias into the hypergradient approximation.
>
> Therefore, the EFIM is the quantity that is consistent with the hypergradient derivation, and is thus the appropriate approximation in the bilevel setting, as motivated by our theoretical results.
>
> **W3&Q2:** We have conducted additional experiments on CIFAR-10 with ResNet-18 (11.2M parameters) as the inner-loop classifier and a UNet augmentation network ( 50K parameters), following the learned data augmentation setup of [9]. This represents a significantly more complex setting than MNIST/FashionMNIST with MLPs: ResNet-18 has 21 Conv2d/Linear layers, and the bilevel problem involves optimizing a generative UNet in the outer loop.
>
> |Method|Mean test|Mean time|
> |-|-|-|
> |Neumann[9] |0.9381 ± 0.0019| 5985 ± 47s|
> |NHGD|0.9384 ± 0.0013|4948 ± 10s|
>
> NHGD achieves equivalent test accuracy to the Neumann series baseline while being **17.3% faster** in wall-clock time.
>
> The efficiency gain comes from our two-GPU parallel pipeline: GPU 0 runs the inner-loop SGD while GPU 1 continuously maintains KFAC factor estimates and pre-computed Kronecker inverses in the background. At each hypergradient step, the pre-computed inverses are applied directly. The measured communication overhead between GPUs is 0.0s (async NVLink transfers fully overlapped with training), and the total wait time for KFAC inversions is under 1s over 200 epochs, showing that the KFAC computation is effectively hidden by the inner-loop training.
>
> These results demonstrate that NHGD scales effectively to standard convolutional architectures and more challenging datasets, maintaining accuracy parity with the Neumann series while providing meaningful wall-clock speedups through hardware-aware parallelism.
>
> (Please see Reply Q7Mb for the full reference list.)

---

> > ### Author Rebuttal · Reviewer_JyLm · 2026-04-03
> >
> > Thanks for the additional experiment. I will I maintain my current positive score.

---

> > > ### Author Response · Authors · 2026-04-04
> > >
> > > Thank you for your constructive review comments and for updating your feedback. We greatly appreciate your time and suggestions, which have helped us improve the paper.

---

### Official Review · Reviewer_Q7Mb · 2026-03-12

**Soundness:** 4
**Presentation:** 4
**Significance:** 3
**Originality:** 4
**Overall Recommendation:** 5
**Confidence:** 3

**Summary:**

**Summary**

This work leverages the statistical structure of the inner optimization problem to propose a fundamentally new method for solving bilevel optimization problems. In particular this is achieved by using the empirical Fisher information matrix as a surrogate for the Hessian.  Such a choice allows for parallel optimization of the hessian inverse along with the inner optimization, providing a clear computational benefit observed in empirics (Figures 2-4) compared to prior methods. Convergence analysis in section 4 is clear to follow and derives a high probability sample complexity bound on the error of the approximate hessian inverse. Experiments on three tasks ( data hyper-cleaning, dataset distillation, and physics informed learning) demonstrate the performance of the proposed method compared to standard baselines.

**Compliance With Llm Reviewing Policy:**

Affirmed.

**Key Questions For Authors:**

1). Could more general inner objectives be considered than the KL Divergence minimization stated in Lines 72-82?

2). Could the same set of metrics be analyzed for each task (e.g., loss, gradient norms, accuracy)? Right now different tasks report different metrics.

**Limitations:**

yes

**Strengths And Weaknesses:**

**Strengths:**
- This work considers a timely and interesting alternative to hyper gradient descent building upon the empirical fisher information matrix of the inner problem. Originality and significance is strong given baselines utilize numerical and algorithmic techniques such as Neumann Series Expansion or Conjugate Gradient.
- Presentation of this work is strong. Problem formulation, motivation, and contributions are clear to follow. Authors outline the convergence analysis in section 4 well highlighting the strength high-probability convergence guarantees provide relative to prior analyses (e.g., Ji et al. 2021).
- Experiments demonstrate significant improvements in computational overhead required for a given level of accuracy and loss.
- This work has many implications for the bilevel optimization community as proposed alternative is general enough to be incorporated across a variety of algorithms, settings, and applications.

**Weaknesses:**
- Limited inner level objectives. Lines 72-82 state the problems consider an important class of problems where inner objective is KL Divergence minimization.
- More comprehensive experiment results. First two tasks report accuracy whereas the final task reports loss.

---

> ### Author Rebuttal · Authors · 2026-03-30
>
> **Q1:**
> Thank you for this insightful comment. We focus on the KL-divergence setting because it provides the Fisher--Hessian equivalence, which makes the Fisher-based approximation both theoretically well justified and computationally tractable.
>
> For more general inner objectives, where the inner problem is not exactly KL-based, the Fisher matrix is still widely used in practice as a surrogate for the Hessian, consistent with the natural-gradient literature. Therefore, we have also implemented our approach in settings where the inner objective is not KL-based, such as PDE-constrained optimization problems, and observed superior empirical performance there as well.
>
> This is in line with the broader observation that natural-gradient methods and their approximations often perform well beyond the exact KL setting across a wide range of machine learning problems [1-4]. Extending the theoretical analysis to these more general settings is an important direction for future work.
>
> **W2&Q2:**
> Thank you for this question. We have conducted additional experiments on CIFAR-10 with ResNet-18 (11.2M parameters) as the inner-loop classifier and a UNet augmentation network ( 50K parameters), following the learned data augmentation setup of [9]. This represents a significantly more complex setting than MNIST/FashionMNIST with MLPs: ResNet-18 has 21 Conv2d/Linear layers, and the bilevel problem involves optimizing a generative UNet in the outer loop.
>
> For hyper-data cleaning and data distillation, we report test accuracy in the main text and the corresponding outer-objective losses in the Appendix E.
>
> For the PDE task, $u_\theta$ is a parameterized PDE solver rather than a classifier, so instead of reporting test accuracy, we report the outer-objective loss and the PDE loss. We therefore report the outer loss in the main text, which measures the discrepancy between the target solution $\hat{u}$ and the learned solution $u_\theta$. We also include the PDE loss of $u_\theta$, which quantifies how well the learned solution satisfies the PDE constraint.
>
> | Method | PDE Loss Mean ± Std |
> |---|---:|
> | NHGD | 0.006907 ± 0.001144 |
> | Broyden | 0.007164 ± 0.001366 |
>
> **References**
> [1] Guzmán-Cordero, Andrés, Felix Dangel, Gil Goldshlager, and Marius Zeinhofer. "Improving Energy Natural Gradient Descent through Woodbury, Momentum, and Randomization." In Advances in Neural Information Processing Systems 38 (NeurIPS 2025), 2025.
>
> [2] Rathore, Pratik, Weimu Lei, Zachary Frangella, Lu Lu, and Madeleine Udell. "Challenges in Training PINNs: A Loss Landscape Perspective." In Proceedings of the 41st International Conference on Machine Learning (ICML), Proceedings of Machine Learning Research, vol. 235, pp. 42159--42191, 2024.
>
> [3] Amari, Shun-Ichi. "Natural gradient works efficiently in learning." Neural computation 10.2 (1998): 251-276.
>
> [4] Martens, James. "New insights and perspectives on the natural gradient method." Journal of Machine Learning Research 21.146(2020): 1-76.
>
> [5] Hao, Z., Ying, C., Su, H., Zhu, J., Song, J., and Cheng, Z. (2023). Bi-level physics-informed neural networks for PDE constrained optimization using Broyden's hypergradients. *In The International Conference on Learning Representations*.
>
> [6] Ji, K., Yang, J., and Liang, Y. Bilevel optimization: Convergence analysis and enhanced design. In International conference on machine learning, pp. 4882–4892. PMLR, 2021.
>
> [7] Arbel, M. and Mairal, J. Amortized implicit differentiation for stochastic bilevel optimization. In International Conference on Learning Representations, 2022.
>
> [8] Hong, M., Wai, H.-T., Wang, Z., and Yang, Z. A twotimescale stochastic algorithm framework for bilevel optimization: Complexity analysis and application to actorcritic. SIAM Journal on Optimization, 33(1):147–180, 2023.
>
> [9] Lorraine, J., Vicol, P. and Duvenaud, D., 2020, June. Optimizing millions of hyperparameters by implicit differentiation. In International conference on artificial intelligence and statistics (pp. 1540-1552). PMLR.
>
> [10] Zhang, Miao, et al. "idarts: Differentiable architecture search with stochastic implicit gradients." International Conference on Machine Learning. PMLR, 2021.
>
> [11] Martens, James. "New insights and perspectives on the natural gradient method." Journal of Machine Learning Research 21.146 (2020): 1-76.
>
> [12] Wainwright, Martin J. High-dimensional statistics: A non-asymptotic viewpoint. Vol. 48. Cambridge university press, 2019.
>
> [13]Madry, Aleksander, et al. "Towards deep learning models resistant to adversarial attacks." arXiv preprint arXiv:1706.06083 (2017).
>
> [14] Ghadimi, Saeed, and Mengdi Wang. "Approximation methods for bilevel programming." arXiv preprint arXiv:1802.02246 (2018).

---

> > ### Author Rebuttal · Reviewer_Q7Mb · 2026-03-31
> >
> > My concerns have been resolved. I maintain my current score of 5: Accept.

---

> > > ### Author Response · Authors · 2026-04-04
> > >
> > > Thank you for your constructive review comments and for updating your feedback. We greatly appreciate your time and suggestions, which have helped us improve the paper.

---

### Decision · Program_Chairs · 2026-04-30

**Decision:**

Accept (regular)

**Comment:**

NHGD replaces the Hessian-inverse bottleneck in bilevel hypergradient estimation with the empirical Fisher, enabling a parallel optimize and approximate scheme with $O(\epsilon^{-2}$ high-probability guarantees. Four reviewers (scores 5, 4, 4, 5) found the contribution original and theoretically sound. I recommend acceptance.